# UDDETTS: Unifying Discrete and Dimensional Emotions for Controllable Emotional Text-to-Speech

## Abstract

Recent large language models (LLMs) have made great progress in the field of text-to-speech (TTS), but they still face major challenges in synthesizing fine-grained emotional speech in an interpretable manner. Traditional methods rely on discrete emotion labels to control emotion categories and intensities, which cannot capture the complexity and continuity of human emotional perception and expression. The lack of large-scale emotional speech datasets with balanced emotion distributions and fine-grained emotional annotations often causes overfitting in synthesis models and impedes effective emotion control. To address these issues, we propose UDDETTS, a universal LLM framework unifying discrete and dimensional emotions for controllable emotional TTS. This model introduces the interpretable Arousal-Dominance-Valence (ADV) space for dimensional emotion description and supports emotion control driven by either discrete emotion labels or nonlinearly quantified ADV values. Furthermore, a semi-supervised training strategy is designed to comprehensively utilize diverse speech datasets with different types of emotional annotations to train the UDDETTS. Experiments show that UDDETTS achieves linear emotion control along three interpretable dimensions, and exhibits superior end-to-end emotional speech synthesis capabilities. Code and demos are available at: `https://anonymous.4open.science/w/UDDETTS`.

## 1 Introduction

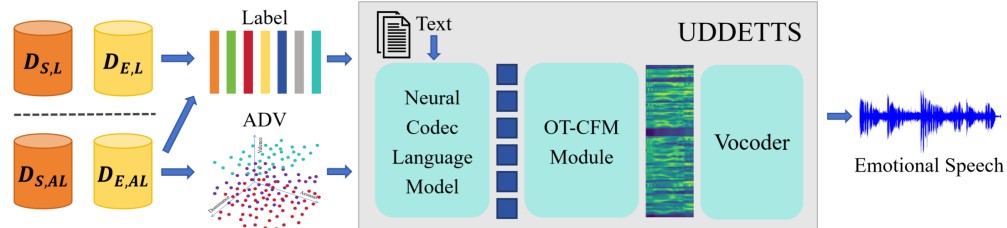

Figure 1: The overview of UDDETTS. It is designed for large-scale emotional speech datasets and integrates discrete label and dimensional ADV annotations to enable controllable emotional TTS.

Recently, a growing number of LLM-based TTS models, e.g. CosyVoice1-3 (Du et al., 2024a;b; 2025), IndexTTS1-2 (Deng et al., 2025; Zhou et al., 2025), FireRedTTS1-2 (Guo et al., 2025; Xie et al., 2025), VibeVoice (Peng et al., 2025), F5-TTS (Chen et al., 2025c), Seed-TTS (Anastassiou et al., 2024), VALL-E (Chen et al., 2025b), Spark-TTS (Wang et al., 2025), have emerged and heralded a new epoch in the field of TTS. These models leverage the strong language understanding of LLMs to generate speech semantic tokens from text tokens, thereby achieving significant advantages in synthesizing expressive speech. In human-computer interaction, enhancing speech expressiveness has become increasingly important, with controllable emotional TTS as a core element. Current LLM-based methods primarily rely on emotion prompts for supervised fine-tuning. They simplify

emotional expression by mapping emotions into predefined discrete categories such as *happy*, *sad*, *angry*, etc. Although some models employ more detailed prompts such as emotion descriptions, timbre, age and prosody to fine-grained control, they do not achieve interpretable disentanglement of speech emotions, so it is still fundamentally constrained by discrete labels in the dataset. Due to the limited variety and granularity of labels and descriptions, this approach generates speech emotions with average expressions per category. In reality, Hong et al. (2025) and Chang (2024) have shown that LLMs can understand complex emotions and exhibit empathy, while Hamann (2012) suggests that emotions exist as a highly interconnected continuum in a continuous space rather than isolated categories. Addressing this limitation requires developing continuous emotion modeling mechanisms in LLM-based TTS models to better capture subtle emotional variations.

With the development of affective computing, dimensional emotion theory (Plutchik, 1980; Russell, 1980; Mehrabian & Russell, 1974; Cowie et al., 2001; Bakker et al., 2014; Gunes & Schuller, 2013) provides a more refined framework for modeling genuine human psychological emotions. Among these, the Arousal-Dominance-Valence (ADV) space (Mehrabian & Russell, 1974) is a commonly used three-dimensional emotion disentanglement space. Arousal represents psychological alertness levels. Low arousal involves being *sleepy* or *bored*, while high arousal involves being *awake* or *excited*. Dominance measures control over others or being controlled, reflecting emotional expression desires. Low dominance involves being *aggrieved* or *weak*, while high dominance involves being *angry* or *amused*. Valence (also known as Pleasure) represents the emotional positivity and negativity, such as being *sad* or *angry* as low valence, while being *happy* or *excited* as high valence. Mehrabian & Russell (1974) and Jia et al. (2025) indicate that these three dimensions account for all variations across 42 emotion scales and cover almost all speech emotion states.

Inspired by the strengths of ADV space in decoupling emotions into interpretable and linearly controllable vectors, how to leverage diverse emotional annotations and address the imbalanced and limited distributions of emotions within the ADV space remains an open challenge. On one hand, existing speech datasets tend to overrepresent neutral emotions, leading to overfitting during training. On the other hand, due to the high cost of emotion annotation, most large-scale emotional speech datasets provide only discrete emotion labels, while only a few offer both discrete labels and dimensional ADV values. This scarcity of ADV annotations leads to low controllable coverage rate in the ADV space. Previous studies (Lugger & Yang, 2008; Wang et al., 2023; Liang et al., 2023) have addressed label-based emotional imbalance. However, none of these methods have explored solutions within the ADV space. Some recent studies (Luo et al., 2025; Li et al., 2025a; Park & Caragea, 2024; Qiu et al., 2024; Lian et al., 2025) have employed semi-supervised training in LLMs to tackle the challenges of diverse annotations. In particular, Luo et al. (2025) shows that semi-supervised training enables interaction across diverse annotation types, and effectively propagates knowledge from labeled to unlabeled data, providing a promising way to address these challenges.

This paper proposes UDDETTS, a universal LLM framework comprising a neural codec language model, an optimal-transport conditional flow matching (OT-CFM) module, and a vocoder, as shown in Figure 1. UDDETTS is the first LLM-based TTS to introduce the interpretable ADV space, enabling fine-grained, decoupled emotion control beyond traditional label-based or description-based methods. It categorizes all datasets into spontaneous emotion datasets and elicited emotion datasets. To address the low controllable coverage rate of the ADV space, it adopts semi-supervised training to accommodate different types of emotional speech datasets, and fuses ADV and label annotations from these datasets. UDDETTS nonlinearly quantizes the ADV space into controllable units as ADV tokens, and introduces an ADV predictor to enhance end-to-end emotional TTS in the absence of emotional annotations. The OT-CFM module employs an emotional mixture encoder to integrate the masked ADV tokens and label token into emotion conditions. We evaluate UDDETTS using objective and subjective metrics across three tasks: label-controlled, ADV-controlled, and end-to-end emotional TTS, comparing it with LLM-based TTS models and analyzing its control performance. Experiments demonstrate UDDETTS achieves more accurate emotional expression while maintaining high naturalness and low WER, and uniquely supports linear control of decoupled emotions along three dimensions.

In summary, our contributions to the community include:

1. We propose UDDETTS, a unified emotional TTS framework that unifies both discrete and dimensional emotions, featuring the first LLM supporting both ADV and label inputs for fine-grained emotional speech synthesis.

2. We propose a nonlinear binning strategy for the ADV space with semi-supervised training to address the imbalance and limited distributions within it, and we leverage large-scale emotional speech datasets to learn a broader range of emotions.

3. UDDETTS disentangles speech emotions in an interpretable manner, enabling linear control along three dimensions, higher naturalness and emotion similarity under label control, and text-adaptive emotion synthesis with text input alone.

## 2 RELATED WORK

Current controllable emotional TTS models can be categorized into label-controlled and space-controlled approaches.

**Label-based control** models learn from discrete emotion categories and intensity levels. For example, current LLM-based models (Du et al., 2024a;b; 2025; Anastassiou et al., 2024; Wang et al., 2025) synthesize emotional speech with specified label prompts, Kang et al. (2023) uses a diffusion model for zero-shot conversion of neutral speech to a target emotional category. To capture fine-grained emotions, Inoue et al. (2024); Liu et al. (2025) employ hierarchical control conditions across coarse and fine granularities. Liu et al. (2024) synthesizes emotional speech based on dialogue context, including emotion labels and intensities. Others explore relative ranking matrices (Zhu et al., 2019), interpolation (Guo et al., 2023), or distance-based quantization (Im et al., 2022) methods to control speech emotional intensity. However, these methods struggle to capture the continuity of emotion distributions.

**Space-based control** models aim to construct a continuous space and capture relationships between different emotions. For example, Li et al. (2025b) proposes a unified TTS framework that learns continuous emotional representation spaces from multimodal emotion prompts. Chen et al. (2023) maps emotions into hyperbolic space to better capture their hierarchical structure. Tang et al. (2023); Zhou et al. (2023); Oh et al. (2023) use interpolation of the embedding space to synthesis speech with a mixture of emotions. AffectEcho (Viswanath et al., 2023) uses a vector quantized space to model fine-grained variations within the same emotion. But these models fail to disentangle the emotion space interpretably, restricting manual control. Recently, EmoSphere-TTS (Cho et al., 2024) and EmoSphere++ (Cho et al., 2025) have explored ADV spaces for interpretable control, using a Cartesian-spherical transformation to control emotion categories and intensities. However, this distorts original emotion clusters and increases overlap, e.g., failing to capture intermediate emotions along the dominance dimension between *angry* and *sad*. Moreover, limited and imbalanced emotional annotations hinder their application to LLMs.

## 3 UDDETTS

UDDETTS needs to learn discrete and dimensional emotions and integrate both in large-scale emotional speech datasets. It categorizes these datasets into spontaneous emotion datasets $\mathbb{D}_S$ and elicited emotion datasets $\mathbb{D}_E$ , and further divides them based on annotation types into four types: $\mathbb{D}_{S,AL}$ ($\mathbb{D}_S$ w/ label & w/ ADV), $\mathbb{D}_{S,L}$ ($\mathbb{D}_S$ w/ label & w/o ADV), $\mathbb{D}_{E,AL}$ ($\mathbb{D}_E$ w/ label & w/ ADV), and $\mathbb{D}_{E,L}$ ($\mathbb{D}_E$ w/ label & w/o ADV). $\mathbb{D}_S$ are recorded in natural scenarios such as conversations, speeches, or performances. In many samples, the emotional representations in speech align with the textual content, enabling the LLM to learn meaningful emotional mappings from a text to ADV and label values. In contrast, $\mathbb{D}_E$ are created by asking speakers to express predefined emotions with varying categories and intensities using the same text. Here, a single text may correspond to multiple labels that do not match its inherent emotion, making it difficult for the LLM to learn emotional mappings from a text to a label, and requiring the ADV or label to guide speech emotions. UDDETTS is designed to control speech emotions using either label or ADV inputs. Its core is a neural codec language model with specially designed token sequences.

### 3.1 SEMI-SUPERVISED NEURAL CODEC LANGUAGE MODEL

#### 3.1.1 MODEL ARCHITECTURE

For the neural codec language model as shown in Figure 2, which is based on the Transformer architecture, the design of input-output sequences is crucial. Inspired by Spark-TTS (Wang et al.,

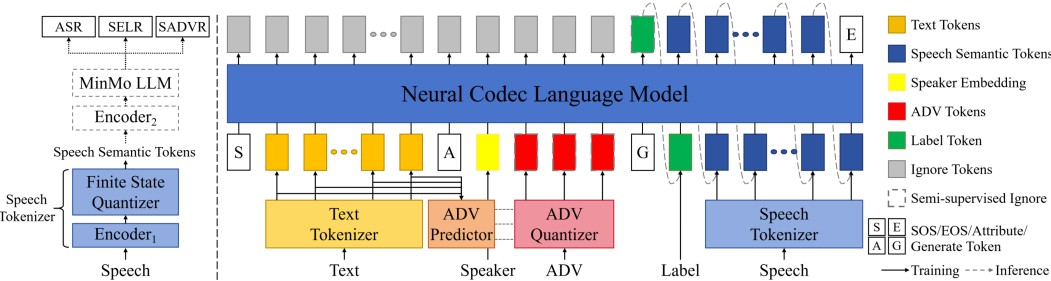

Figure 2: Left: the supervised multi-task speech tokenizer. Right: the neural codec language model running autoregressively until EOS. During semi-supervised training, ADV tokens in the input and label token in the output are dynamically masked depending on dataset type.

2025), the LLM separates textual content from speech attribute features, further decoupling speaker timbre from emotional representations within the latter. It integrates the input-output sequences of different dataset types into a unified model, as defined in Eqs. (1-3).

$$\mathbb{D}_{S|E,AL}: \quad \begin{aligned} \boldsymbol{x}_{\text{input}} &= [x_{\text{sos}}, \boldsymbol{x}_{\text{text}}, x_{\text{attr}}, x_{\text{spk}}, \boldsymbol{x}_{\text{adv}} \in \mathbb{Z}^3_{[1,m]}, x_{\text{gen}}, x_{\text{lbl}} \in \mathbb{Z}^1_{[0,n]}, \boldsymbol{x}_{\text{sem}}] \\ \boldsymbol{x}_{\text{output}} &= [\boldsymbol{x}_{\text{ign}}, x_{\text{lbl}} \in \mathbb{Z}^1_{[1,n]}, \boldsymbol{x}_{\text{sem}}, x_{\text{eos}}] \end{aligned} \tag{1}$$

$$\mathbb{D}_{S,L}: \quad \begin{aligned} \boldsymbol{x}_{\text{input}} &= [x_{\text{sos}}, \boldsymbol{x}_{\text{text}}, x_{\text{attr}}, x_{\text{spk}}, \boldsymbol{x}_{\text{ign}} \in \mathbb{Z}^3, x_{\text{gen}}, x_{\text{lbl}} \in \mathbb{Z}^1_{[0,n]}, \boldsymbol{x}_{\text{sem}}] \\ \boldsymbol{x}_{\text{output}} &= [\boldsymbol{x}_{\text{ign}}, x_{\text{lbl}} \in \mathbb{Z}^1_{[1,n]}, \boldsymbol{x}_{\text{sem}}, x_{\text{eos}}] \end{aligned} \tag{2}$$

$$\mathbb{D}_{E,L}: \quad \begin{aligned} \boldsymbol{x}_{\text{input}} &= [x_{\text{sos}}, \boldsymbol{x}_{\text{text}}, x_{\text{attr}}, x_{\text{spk}}, \boldsymbol{x}_{\text{ign}} \in \mathbb{Z}^3, x_{\text{gen}}, x_{\text{lbl}} \in \mathbb{Z}^1_{[0,n]}, \boldsymbol{x}_{\text{sem}}] \\ \boldsymbol{x}_{\text{output}} &= [\boldsymbol{x}_{\text{ign}}, x_{\text{ign}} \in \mathbb{Z}^1, \boldsymbol{x}_{\text{sem}}, x_{\text{eos}}] \end{aligned} \tag{3}$$

where $\boldsymbol{x}_{\text{input}}$ and $\boldsymbol{x}_{\text{output}}$ are the input sequence and output sequence of the neural codec language model. Specifically, $x_{\text{sos}}$, $x_{\text{eos}}$, $x_{\text{attr}}$ and $x_{\text{gen}}$ represent the start-of-sequence token, end-of-sequence token, attribute-start token, and generation-start token, respectively. All of them are fixed values and belong to $\mathbb{Z}^1$. $\boldsymbol{x}_{\text{ign}}$ is the ignore tokens, used to mask positions in the $\boldsymbol{x}_{\text{output}}$ during training. $x_{\text{text}}$ is obtained by processing raw text with a Byte Pair Encoding (BPE)-based tokenizer (Radford et al., 2023). To align semantic information, $\boldsymbol{x}_{\text{text}}$ is encoded into text embeddings via a Conformer-based text encoder. $x_{\text{spk}}$ is the speaker id, encoded as the speaker embedding computed by averaging timbre vectors extracted from all *neutral* emotional speech samples of this speaker using a voiceprint model (Chen et al., 2024). This embedding captures speaker timbre while excluding emotional representations. $\boldsymbol{x}_{\text{adv}}$ is obtained from ADV values using an ADV quantizer based on the nonlinear binning described in Section 3.1.2, and $m$ is the number of bins along each dimension. $x_{\text{lbl}}$ is the emotion label token, and $n$ is the number of label token types. $\boldsymbol{x}_{\text{sem}}$ is the speech semantic tokens enriched with emotional representations, extracted by a novel speech tokenizer shown in Figure 2.

To ensure that $x_{\text{sem}}$ captures rich paralinguistic emotional information, we design a supervised multi-task speech tokenizer inspired by CosyVoice3 (Du et al., 2025). Specifically, the Finite Scalar Quantization (FSQ) module (Mentzer et al., 2024) is inserted into the encoder of the MinMo model (Chen et al., 2025a), which is then jointly trained on automatic speech recognition (ASR), speech emotion label recognition (SELR), and speech ADV recognition (SADVR).

### 3.1.2 EMOTION QUANTIFICATION

In the ADV space, emotions are continuously distributed. For controllability, these continuous vectors are quantized into tokens $\boldsymbol{x}_{\text{adv}} = [x_{\text{a}}, x_{\text{d}}, x_{\text{v}}] \in \mathbb{Z}^3_{[1,m]}$, where $x_{\text{a}}$ (arousal) controls the intensity of the emotion provoked by a stimulus, $x_{\text{d}}$ (dominance) controls the level of control exerted by the stimulus, and $x_{\text{v}}$ (valence) controls the positivity or negativity of an emotion. However, due to imbalanced emotion distributions and limited ADV values in these datasets, the distributions along the three dimensions exhibit approximately normal patterns, and certain regions of the ADV space remain underrepresented, as shown in Appendix E. To address these problems, we design an ADV quantizer by exploring different nonlinear binning algorithms (Garca et al., 2016) for each of the three dimensions, and finally select the clustering-based binning algorithm to balance uniformity

and discriminability. Then, to balance control granularity and linearity, the ADV quantizer uses the central limit theorem (Punhani et al., 2022) to determine the number of bins. Details of the nonlinear binning algorithm derivation are given in the Appendix E.

We observe that different emotion labels generally form distinct clusters in the ADV space, as shown in in Appendix D. However, some labels show substantial overlap, indicating ambiguity in their emotional boundaries. So we unify semantically similar emotion labels in the datasets into a single token. For example, both *happy*, *amused* and *laughing* are grouped under the *happy* category and assigned the same token.

### 3.1.3  ADV PREDICTOR

We also observe that without control conditions, predicting $x_{\text{lbl}}$ and $\boldsymbol{x}_{\text{sem}}$ solely from $\boldsymbol{x}_{\text{text}}$ performs poorly, often yielding speech with *neutral* emotion. To enhance end-to-end emotional TTS, we introduce an ADV predictor that first estimates pseudo-ADV $\boldsymbol{adv}_{\text{pred}}$ from $\boldsymbol{x}_{\text{text}}$, $\boldsymbol{adv}_{\text{pred}}$ are then quantized by the ADV quantizer into pseudo-ADV tokens $\boldsymbol{x}_{\text{adv}_{\text{pred}}}$, which are fed into the neural codec language model together with $\boldsymbol{x}_{\text{text}}$. The ADV predictor, inspired by Park et al. (2021); Wen et al. (2021), employs a RoBERTa encoder followed by softmax and norm layers over the pooled output. It is trained jointly with the LLM and loss function is defined as:

$$\mathcal{L}_{ADV} = \sum_{x \in \{a,d,v\}} \alpha ||\boldsymbol{x}_{\text{pred}} - \boldsymbol{x}_{\text{true}}||^2 + \sum_{b=1}^{B} ||\boldsymbol{adv}_{\text{pred}_b} - \boldsymbol{c}_b||^2, \tag{4}$$

the first term computes the MSE of $\boldsymbol{adv}_{\text{pred}}$ across three decoupled dimensions, while the second term minimizes its distance to the bin center $\boldsymbol{c}_b$ of $\boldsymbol{adv}_{\text{true}}$ for each sample.

### 3.1.4  TRAINING AND INFERENCE

During training, due to the mixture of datasets, each batch may include samples from multiple sources. For samples with $\boldsymbol{x}_{\text{adv}} \neq \boldsymbol{x}_{\text{ign}}$ in a batch (i.e., from $\mathbb{D}_{S,AL}$ or $\mathbb{D}_{E,AL}$, see Eq. 1), their corresponding $x_{\text{lbl}}$ in $\boldsymbol{x}_{\text{output}}$ can be correctly predicted from the text and ADV, and is therefore not masked. For samples where $\boldsymbol{x}_{\text{adv}} = \boldsymbol{x}_{\text{ign}}$, the masking depends on the dataset type: if the sample comes from $\mathbb{D}_{S,L}$, $x_{\text{lbl}}$ in $\boldsymbol{x}_{\text{output}}$ is not masked, since the text emotion and label are consistent; but if the sample comes from $\mathbb{D}_{E,L}$, $x_{\text{lbl}}$ in $\boldsymbol{x}_{\text{output}}$ needs to be masked. In spontaneous emotional datasets $\mathbb{D}_S$, many samples exhibit ambiguous emotional expressions and are labeled as *Unknown* (see Table 6 in the Appendix). When $x_{\text{lbl}} = 0$ in $x_{\text{input}}$, the corresponding $x_{\text{lbl}}$ in $x_{\text{output}}$ is masked during training. We design a label token position-aware smoothing loss function for semi-supervised training, as defined in follow Eqs. (5,6):

$$\mathcal{L}_{LLM} = -\frac{1}{L+2} \sum_{l=1}^{L+2} w_{\text{emo}}(l) p(v_l) \log q(v_l) + \mathcal{L}_{ADV}, \tag{5}$$

$$\text{where} \quad p(v_l) = \begin{cases} 1 - \epsilon, & \text{if } v_l = \mu_l \\ \frac{\epsilon}{K}, & \text{if } v_l \neq \mu_l \end{cases}, \quad w_{\text{emo}}(l) = \begin{cases} 0, & \text{if } \mu_l = x_{\text{lbl}} = x_{\text{ign}} \text{ or } 0 \\ 5.0, & \text{if } \mu_l = x_{\text{lbl}} \neq x_{\text{ign}} \text{ or } 0 , \\ 1.0, & \text{otherwise} \end{cases} \tag{6}$$

here, $L + 2$ is the length of $\boldsymbol{x}_{\text{loss}} = [x_{\text{lbl}}, \boldsymbol{x}_{\text{sem}}, x_{\text{eos}}]$ in $\boldsymbol{x}_{\text{output}}$. $v_l$ and $\mu_l$ denote the predicted token and the ground-truth token at position $l$ in $\boldsymbol{x}_{\text{loss}}$. $w_{\text{emo}}(l)$ is the position-dependent weighting scale. When the $x_{\text{lbl}}$ is $x_{\text{ign}}$ or 0, indicating that the sample belongs to $\mathbb{D}_{E,L}$ or the label is *Unknown* — the loss at $x_{\text{lbl}}$ position is masked. Otherwise, the loss at $x_{\text{lbl}}$ position is up-weighted to accelerate convergence. $p(v_l)$ is used for label smoothing, where $K$ is the vocabulary size and $\epsilon$ is a small smoothing parameter.

During inference, the LLM operates in three modes, corresponding to three different tasks:

1. The first task controls speech emotion categories using a label: it uses $\boldsymbol{x}_{\text{text}}$ and $x_{\text{lbl}}$, with the $\boldsymbol{x}_{\text{adv}}$ ignored, to generate label-conditioned $\boldsymbol{x}_{\text{sem}}$.

2. The second task controls fine-grained emotions using ADV tokens: it uses $\boldsymbol{x}_{\text{text}}$ and $\boldsymbol{x}_{\text{adv}}$ to predict $x_{\text{lbl}}$ and then generates $\boldsymbol{x}_{\text{sem}}$ autoregressively.

3. The third task predicts text-adaptive emotions directly from texts: it it uses only $\boldsymbol{x}_{\text{text}}$ to predict $\boldsymbol{x}_{\text{sem}}$, while $\boldsymbol{x}_{\text{adv}_{\text{pred}}}$ serve as intermediate tokens predicted from the text.

### 3.2 SEMI-SUPERVISED CONDITIONAL FLOW MATCHING

To synthesize emotional speech, UDDETTS reconstructs the speech semantic tokens $x_{\text{sem}}$ into mel-spectrograms via an OT-CFM module. This module is conditioned on the speaker embedding $\boldsymbol{E}_{\text{spk}}$, the semantic embedding $\boldsymbol{E}_{\text{sem}}$ and the emotion conditions $\boldsymbol{E}_{\text{emo}}$. Here, $\boldsymbol{E}_{\text{sem}}$ is obtained by encoding the generated $x_{\text{sem}}$ via a Conformer-based semantic encoder, while $\boldsymbol{E}_{\text{emo}}$ is derived from both $x_{\text{lbl}}$ and $x_{\text{adv}}$ to enhance the emotional guidance of the synthesized speech.

To generate $\boldsymbol{E}_{\text{emo}}$, the OT-CFM module employs an emotional mixture encoder, as illustrated in Figure 3. This encoder fuses the masked $x_{\text{lbl}}$ and $x_{\text{adv}}$. Specifically, the ADV encoder first encodes $x_{\text{a}}$, $x_{\text{d}}$ and $x_{\text{v}}$ separately into $\boldsymbol{E}_{\text{a}}$, $\boldsymbol{E}_{\text{d}}$ and $\boldsymbol{E}_{\text{v}}$, which are then concatenated and passed through an interaction layer to obtain the ADV embedding $\boldsymbol{E}_{\text{adv}}$. The label encoder directly encodes $x_{\text{lbl}}$ into a label embedding $\boldsymbol{E}_{\text{lbl}}$. A multi-head attention layer is applied, using $\boldsymbol{E}_{\text{lbl}}$ as the query and $\boldsymbol{E}_{\text{adv}}$ as the key and value. resulting in a label-attended emotion embedding $\boldsymbol{E}_{\text{emo}}^{\text{attn}}$. Finally, a gate layer combined with the semi-supervised gating algorithm described in Eq. (7) produces the final emotion conditions $\boldsymbol{E}_{\text{emo}}$.

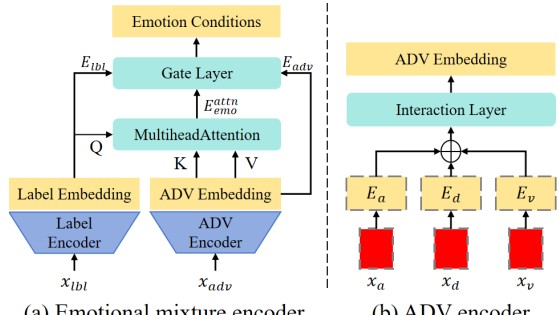

Figure 3: The emotional mixture encoder of OT-CFM module to generate the emotion conditions.

$$\boldsymbol{E}_{\text{emo}} = \begin{cases} \boldsymbol{E}_{\text{adv}} & \text{if } x_{\text{lbl}} = 0 \\ (gate + 1) \cdot \boldsymbol{E}_{\text{lbl}} & \text{if } x_{\text{lbl}} \neq 0 \text{ and } \boldsymbol{x}_{\text{adv}} = \boldsymbol{x}_{\text{ign}} \\ gate \cdot \boldsymbol{E}_{\text{lbl}} + (1 - gate) \cdot \boldsymbol{E}_{\text{emo}}^{\text{attn}} & \text{if } x_{\text{lbl}} \neq 0 \text{ and } \boldsymbol{x}_{\text{adv}} \neq \boldsymbol{x}_{\text{ign}} \end{cases} \quad (7)$$

The OT-CFM module defines a time-dependent vector field $\boldsymbol{v}_t(\boldsymbol{X}) : [0, 1] \times \mathbb{R}^{L \times D} \to \mathbb{R}^{L \times D}$, and uses an ordinary differential equation (Onken et al., 2021) to find the optimal-transport (OT) flow $\phi_t^{OT}$. All condition, including $\boldsymbol{E}_{\text{spk}}$, $\boldsymbol{E}_{\text{sem}}$ and $\boldsymbol{E}_{\text{emo}}$, are fed into a U-net neural network $\mathbf{U}_\theta$ to match the vector field $\boldsymbol{v}_t(\boldsymbol{X})$ to $\boldsymbol{w}_t(\boldsymbol{X})$ with learnable parameters $\theta$:

$$\boldsymbol{v}_t(\phi_t^{OT}(\boldsymbol{X}_0, \boldsymbol{X}_1)|\theta) = \mathbf{U}_\theta(\phi_t^{OT}(\boldsymbol{X}_0, \boldsymbol{X}_1), \boldsymbol{E}_{\text{spk}}, \boldsymbol{E}_{\text{sem}}, \boldsymbol{E}_{\text{emo}}, \boldsymbol{t}), \quad (8)$$

$$\boldsymbol{w}_t(\phi_t^{OT}(\boldsymbol{X}_0, \boldsymbol{X}_1)|\boldsymbol{X}_1) = \boldsymbol{X}_1 - (1 - \sigma)\boldsymbol{X}_0, \quad (9)$$

where $\boldsymbol{X}_0 \sim \mathcal{N}(0, \tau^{-1}\boldsymbol{I})$, $\boldsymbol{X}_1$ is a learned approximation of the mel-spectrogram distributions, $\boldsymbol{t}$ is the timestep using a cosine schedule (Nichol & Dhariwal, 2021) to prevent rapid noise accumulation from linear addition. The conditional flow matching loss function is shown in Eq. (10):

$$\mathcal{L}_{CFM} = \mathbb{E}_{\boldsymbol{X}_0, \boldsymbol{X}_1} ||\boldsymbol{w}_t(\phi_t^{OT}(\boldsymbol{X}_0, \boldsymbol{X}_1)|\boldsymbol{X}_1) - \boldsymbol{v}_t(\phi_t^{OT}(\boldsymbol{X}_0, \boldsymbol{X}_1)|\theta)||^2. \quad (10)$$

During inference, $x_{\text{lbl}}$ is derived directly from the input in the first task (1), while in the second and third tasks (2, 3), $x_{\text{lbl}}$ is obtained from the label predicted by the LLM.

## 4 EXPERIMENTS

### 4.1 DATASETS

To evaluate the UDDETTS model, we collect large-scale English emotional speech datasets, including MSP (Lotfian & Busso, 2019), IEMOCAP (Busso et al., 2008), MELD (Poria et al., 2019), MEAD(Wang et al., 2020), CMU-MOSEI (Bagher Z et al., 2018), ESD (Zhou et al., 2022), EmoV-DB (Adigwe et al., 2018), Expresso (Nguyen et al., 2023), CREMA-D (Cao et al., 2014), RAVDESS (Livingstone et al., 2018), EmoTale (Hjuler et al., 2025), EU-Emotion (Lassalle et al., 2018) and 118.6 hours of internally annotated emotional speech. Each dataset is annotated with either emotion labels or ADV values. We also leverage 49400+ hours English general speech datasets w/o emotional annotations to support early-stage TTS training. All samples undergo preprocessing: emotion labels, punctuation, numbers, and other special characters are standardized; ADV values are normalized to [1,7]; annotation errors are removed; and speech recordings are resampled to 16 kHz. We

Table 1: Comparison of subjective and objective evaluation results across LLM-based TTS models.

| Models | MOS↑ | $P_m$↑ | $R_m$↑ | UTMOS↑ | WER(%)↓ | SS↑ | ES↑ | STOI↑ | PESQ-WB↑ |
|---|---|---|---|---|---|---|---|---|---|
| UDDETTS | 4.29±0.12 | **0.94** | **0.90** | 4.25 | 2.40 | 0.702 | **0.833** | 0.90 | **2.80** |
| CosyVoice | 4.02±0.08 | 0.83 | 0.73 | 3.87 | 4.35 | 0.679 | 0.635 | 0.83 | 2.16 |
| CosyVoice2 | 4.20±0.10 | 0.85 | 0.75 | 4.10 | 2.42 | 0.733 | 0.720 | 0.88 | 2.59 |
| CosyVoice3 | **4.35±0.10** | 0.85 | 0.82 | **4.48** | **1.45** | 0.784 | 0.790 | 0.92 | 2.68 |
| IndexTTS | 4.20±0.15 | 0.83 | 0.72 | 3.95 | 2.45 | 0.715 | 0.678 | 0.89 | 2.43 |
| IndexTTS2 | 4.29±0.10 | 0.87 | 0.80 | 4.20 | 1.69 | **0.792** | 0.778 | **0.94** | 2.60 |
| FireRedTTS | 3.95±0.07 | 0.74 | 0.65 | 3.80 | 3.85 | 0.635 | 0.605 | 0.86 | 2.30 |
| FireRedTTS2 | 4.15±0.06 | 0.83 | 0.76 | 3.85 | 3.19 | 0.684 | 0.723 | 0.90 | 2.50 |
| Spark-TTS | 4.18±0.13 | 0.85 | 0.77 | 4.04 | 2.03 | 0.678 | 0.680 | 0.89 | 2.46 |
| F5-TTS | 4.18±0.07 | 0.88 | 0.78 | 4.30 | 1.82 | 0.723 | 0.709 | 0.92 | 2.35 |
| VALL-E | 3.79±0.15 | 0.62 | 0.69 | 3.58 | 5.98 | 0.590 | 0.594 | 0.81 | 1.91 |
| CosyVoice + ADV | 4.15±0.05 | 0.90 | 0.81 | 4.10 | 4.08 | 0.680 | 0.815 | 0.86 | 2.66 |
| UDDETTS w/o EME | 4.20±0.10 | 0.90 | 0.87 | 4.18 | 2.35 | 0.682 | 0.820 | 0.90 | 2.71 |

remove samples with overlapping speakers, instrumental music, excessive noise, other languages, missing transcriptions, and durations longer than 30 seconds. To reduce speaker timbre confusion, we remove samples from *Unknown* speakers and discard speakers with fewer than four utterances. Appendix D summarizes the statistics of collected datasets after cleaning. In total, 19 emotion labels are used, with corresponding label tokens [0, 9] and sample counts listed in Table 6 in Appendix.

## 4.2 IMPLEMENTATION DETAILS

We first train the speech tokenizer on the full training set, which converges within 500k steps. The trained tokenizer is then used to extract speech semantic tokens. For UDDETTS, the first stage involves training LLM-0.70B and OT-CFM-0.35B on English speech corpora without emotional annotations, with a peak learning rate of 1e-3, 5000 warm-up steps, and 15 epochs until convergence. In the second stage, we perform semi-supervised training on large-scale English emotional speech datasets, with the text encoder frozen, a peak learning rate of 1e-4 and 2500 warm-up steps. UD-DETTS converges within 30 epochs. The generated mel-spectrograms are converted into emotional speech using a HiFi-GAN (Kong et al., 2020) vocoder, fine-tuned on our datasets for 5 epochs. All UDDETTS training is conducted on 24 NVIDIA A800-80GB GPUs with 64-core CPUs, using the Adam optimizer, gradient accumulation of 2, and a maximum total frame length of 5000 per batch. For evaluation, we collect and design a text corpus as the test set, as shown in Appendix F.

## 4.3 LABEL-CONTROLLED EMOTIONAL TTS

To evaluate label-controlled synthesis, each *neutral* text is paired with five emotions (*neutral*, *happy*, *angry*, *disgust*, and *sleepiness*), whose training sample sizes decrease stepwise, and used as control inputs for UDDETTS under the first task (1). We compare different LLM-based TTS models, as shown in Appendix C, under label prompts (e.g., "Angry<|endofprompt|>Content Text"). We conduct both subjective and objective evaluations of the synthesized speech. Subjective evaluation involves 12 participants, measuring 5-point Mean Opinion Scores (MOS) for speech naturalness and a five-class emotion confusion matrix to assess the robustness of label-based control, from which macro-Precision $P_m$ and macro-Recall $R_m$ are computed. Objective evaluation uses Whisper-large-v3 model (Radford et al., 2023) for Word Error Rate (WER) to assess speech intelligibility, 3D-Speaker speaker verification model (Chen et al., 2024) for Speaker Similarity (SS), emotion2vec[1] model for Emotion Similarity (ES), speechmetrics[2] to calculate Short-Time Objective Intelligibility (STOI) and Perceptual Evaluation of Speech Quality - Wideband (PESQ-WB), and SpeechMOS[3] to calculate UTMOS. The results in Table 1 show that UDDETTS achieves higher naturalness of synthesized speech compared with CosyVoice1-2, IndexTTS, FireRedTTS1-2, Spark-TTS, F5-TTS, and VALL-E, while also maintaining low WER and high speaker similarity. More importantly, UDDETTS obtains the highest scores in both $P_m$ and $R_m$ of the emotion confusion matrix, as well as in Emotion Similarity and PESQ-WB. We further integrate the proposed ADV framework—including ADV tokens, quantizer, predictor, speech tokenizer, and emotional mixture

---

[1]https://github.com/ddlBoJack/emotion2vec

[2]https://github.com/aliutkus/speechmetrics

[3]https://github.com/tarepan/SpeechMOS

Table 2: Subjective evaluation results of linear emotion control along the three ADV dimensions. The right side *Linear Binning* presents the results of ablation experiments.

| Dimension | Range | Nonlinear Binning | | Linear Binning | |
|---|---|---|---|---|---|
| | | SRC | KW | SRC | KW |
| Arousal | [1-14, 7, 7] | 0.85 | 0.70 | 0.52 | 0.48 |
| Dominance | [14, 1-14, 1] | 0.78 | 0.68 | 0.48 | 0.50 |
| Valence | [14, 14, 1-14] | 0.92 | 0.83 | 0.57 | 0.58 |

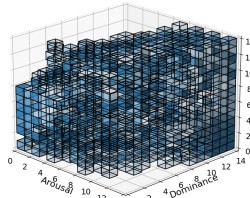 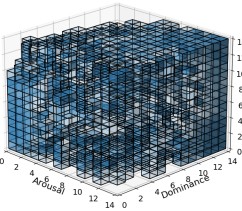 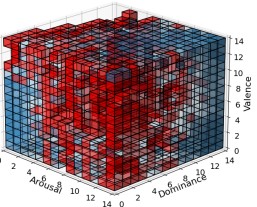 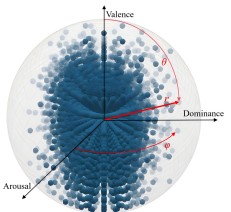

Figure 4: ADV space with 14×14×14 controllable units: **linear binning (60.83%)** vs. **nonlinear binning (77.89%)** vs. **after semi-supervised training (89.35%)**, and **spherical coordinate system (55.40%)**. Color opacity positively correlates with the sample density within each controllable unit.

encoder—into CosyVoice1 by fine-tuning the CosyVoice-300M on our English emotional speech datasets. Results in Table 1 show that CosyVoice + ADV outperforms the original CosyVoice in both speech naturalness and label-based emotion control accuracy. These results indicate that UDDETTS achieves higher accuracy and demonstrates stronger robustness in label-controlled emotional TTS.

## 4.4 ADV-CONTROLLED EMOTIONAL TTS

To evaluate UDDETTS's ability to linearly control emotions along each of three ADV dimensions, we conduct experiments on the second task (2) by adjusting the values of $x_{adv}$ to control speech emotions. Using the nonlinear binning algorithm with $m = 14$ bins, UDDETTS quantizes the ADV values $adv \in \mathbb{R}^3$ into controllable units $x_{adv} \in \mathbb{Z}^3_{[1,14]}$. In each experiment, two dimensions are fixed while the third is varied, yielding three test settings: Arousal test $x_{adv}$=[1-14, 7, 7], Dominance test under strong negative emotions $x_{adv}$=[14, 1-14, 1], Valence test under strong expressiveness $x_{adv}$=[14, 14, 1-14]. Stronger emotions are assumed to exhibit greater perceptual separability during ranking. For each test, we synthesize 14 speech samples from 10 *neutral* texts drawn from the text corpus and ask participants to rank them using the SAM system as shown in Appendix G. We use Spearman's Rank Correlation (SRC) to evaluate the alignment between each participant's rankings and ground-truth rankings, and report the average score. And Kendall's W (KW) is used to evaluate inter-rater agreement across 12 participants:

$$SRC = 1 - \frac{6 \sum d_i^2}{n(n^2 - 1)}, \quad KW = \frac{12S}{k^2(n^3 - n)}, \quad (11)$$

where $d_i$ is the rank difference between two rankings, $n$ is the number of samples, $S$ is the variance of rank sums, and $k$ is 12. As shown in Table 2, SRC values near 1.0 show that perceived emotions change linearly with the nonlinearly binned $x_{adv}$. The KW scores above 0.6 reflect strong inter-rater agreement, confirming the reliability of the results.

To objectively validate the robustness of the nonlinear binning algorithm, we conduct a sensitivity analysis on the number of bins $m$ using features extracted by emotion2vec[1]. As shown in Figure 5, emotional control maintains linearity at $m = 14$, achieving an optimal balance between control granularity and linearity. When $m < 12$, emotional transitions become rapid and non-smooth at extreme ADV values with reduced granularity. When $m > 16$, insufficient samples per control unit lead to poor generalization capability and deviations in intermediate emotional transitions.

As shown in Figure 4, nonlinear binning and semi-supervised training significantly expand the controllable coverage of the ADV space. Nonlinear binning produces more uniformly distributed controllable units than linear binning, increasing the coverage rate from 60.83% to 77.89%. while

Table 3: Subjective preference (%) test and UTMOS results on ADV-controlled mixed emotions.

| Mixed emotion | UDDETTS | UTMOS | Similar | EmoSphere++ | UTMOS | p-value |
|---|---|---|---|---|---|---|
| angry-sad | **74.50** | 4.35 | 20.00 | 5.50 | 4.03 | 0.001 |
| sleepiness-sad | **52.40** | 3.96 | 28.35 | 19.25 | 3.85 | 0.019 |
| happy-surprise | **60.78** | 4.28 | 23.42 | 15.80 | 3.97 | 0.010 |
| disgust-angry | **43.33** | 4.18 | 33.34 | 23.33 | 3.90 | 0.032 |

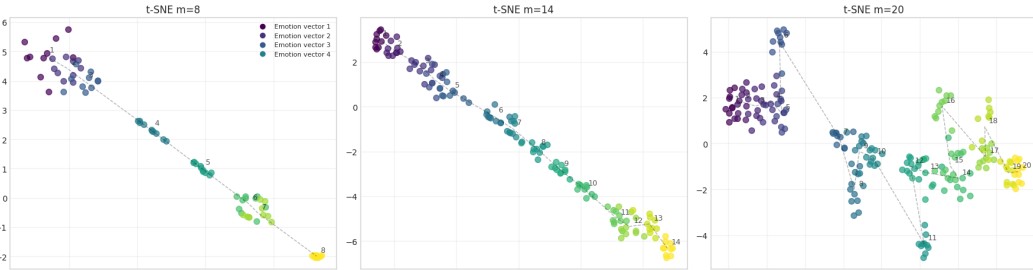

Figure 5: Sensitivity analysis of bin count $m$ on granularity and linearity, with t-SNE visualization of extracted emotion vectors. Each color represents a set of emotion vector samples extracted from speech synthesized under linearly transformed ADV control.

semi-supervised training further raises it to 89.35%. Red regions in the ADV space highlight areas capable of synthesizing emotional speech corresponding to unseen ADV values. For example, at $x_{adv}$=[14, 1, 1], where no training samples exist, the model can still synthesize reasonable *sobbing-like* speech. This indicates that semi-supervised training promotes the transfer of label knowledge to the ADV space, thereby enabling broader and finer-grained control of emotional TTS.

We also evaluate the robustness of UDDETTS when the label or ADV inputs fall outside the label and ADV ranges of the training set, as shown in Appendix H. To further evaluate the influence of each ADV dimension on emotion expression, we also analyze the relationship between ADV and prosodic features of speech, as detailed in the Appendix I. Together, these results demonstrate that UDDETTS achieves fine-grained, interpretable, and linear emotion control along three psychological dimensions, surpassing the capabilities of traditional label-based methods.

Additionally, to evaluate the performance of UDDETTS in capturing intermediate emotions against the traditional non-LLM-based EmoSphere++ (Cho et al., 2025), a model that employs an emotion-adaptive spherical vector (EASV) within a spherical coordinate system where angles control emotion style and the radius controls intensity, we plot the ADV space of both methods on the same emotional speech dataset. As shown in Figure 4, UDDETTS (89.35%) exhibits broader coverage compared to EmoSphere++ (55.40%), with smoother color gradients in control units, indicating a more uniform sample distribution. We further compare the emotional control accuracy using equivalent ADV values for four intermediate emotions: angry-sad, sleepiness-sad, happy-surprise, and disgust-angry. The ADV values are set to the medians between the centers of the corresponding emotion clusters in Figure 6. ABX test results in Table 3 show that participants significantly prefer UDDETTS-synthesized intermediate emotions, particularly for angry-sad, demonstrating its superior ability to mitigate emotional regional overfitting and capture mixed emotions effectively.

## 4.5 END-TO-END EMOTIONAL TTS

To evaluate the UDDETTS's ability for text-adaptive emotion synthesis using text input alone, we conduct experiments on the third task (3) and select the text corpus featuring diverse and explicit emotional attributes (see Appendix F). We compare UDDETTS with two description-based baselines, providing each baseline with a neutral reference speech, the target text, and a natural language description (e.g. "Synthesize the emotional speech that best matches the text<|endofprompt|>Content Text"). A subjective preference (%) test involving 12 participants is conducted to evaluate which model generates speech with more appropriate emotions, with the

Table 4: Subjective preference (%) test results on end-to-end emotional TTS.

| UDDETTS | Similar | CosyVoice2 | p-value | UDDETTS | Similar | IndexTTS2 | p-value |
|---|---|---|---|---|---|---|---|
| **67.33** | 19.45 | 13.22 | 0.001 | **58.60** | 29.23 | 12.17 | 0.012 |
| w/o ADV predictor | Similar | CosyVoice2 | p-value | w/o ADV predictor | Similar | IndexTTS2 | p-value |
| **46.88** | 24.30 | 28.82 | 0.035 | 28.50 | **50.40** | 21.10 | 0.104 |

p-value of t-test used to assess significance of differences. As shown in Table 4, participants demonstrate a clear preference for UDDETTS ($p < 0.05$).

To quantify the emotional consistency between text semantics and synthesized speech, and to validate the ADV predictor (trained with an RMSE of 1.25), we extract pseudo-ADV values from intermediate text predictions and speech-based ADV values through the SADVR task using the multi-task speech tokenizer (trained with an RMSE of 0.68). These text-derived and speech-derived ADV values are visualized in the ADV space, as shown in Figure 8 in Appendix F. The close alignment between them demonstrates that UDDETTS effectively maps fine-grained emotional representations from text to speech.

Overall, these results confirm that UDDETTS exhibits superior end-to-end capabilities in text-adaptive emotion understanding and emotional TTS.

### 4.6 ABLATION STUDIES

We conduct four ablation studies to evaluate the effectiveness of key components in UDDETTS. First, removing the ADV predictor in the third task biases the synthesized speech toward neutral and lowers the scores, as shown in Table 4, indicating that the pseudo-ADV predicted by the ADV predictor helps the LLM capture intrinsic emotions from the text. Second, we remove the emotional mixture encoder and $E_{\text{emo}}$ from the OT-CFM module and rely solely on $E_{\text{sem}}$ to reconstruct mel-spectrograms. This modification leads to a reduction in emotional expressiveness, as seen in the last row (w/o EME) of Table 1. Third, we replace nonlinear binning algorithm in the ADV quantizer with a linear one. Both SRC and KW scores drop significantly in Table 2, indicating that imbalanced emotion distributions lead the model to overfit dense ADV regions, thereby impeding linear control. Finally, training only on $\mathbb{D}_{S,AL}$ without semi-supervised learning reduces the controllable coverage rate of the ADV space to 70%, and fails to synthesize the *sobbing-like* emotion at $x_{\text{adv}}$=[14, 1, 1] and other unseen emotions. This highlights the pivotal role of unlabeled ADV data in transferring discrete emotion knowledge into the ADV space and expanding control coverage.

## 5 LIMITATIONS AND FUTURE WORK

The performance of UDDETTS is limited by the quality of ADV annotations in the datasets. Subjective variation among annotators can produce inconsistent ADV labels, which negatively impacts the model's ability for linear emotional control; increasing dataset size and selecting samples with consistent annotations are the primary way to mitigate this issue. Additionally, for texts with ambiguous emotional attributes, the ADV predictor often struggles to infer appropriate ADV values. Since the same text can express different emotions in different contexts, incorporating multimodal information is necessary for more accurate emotion understanding. In future work, we plan to extract emotional representations from multimodal sources and dialogue context, mapping them into the ADV space to better capture emotions.

## 6 CONCLUSION

In this paper, we introduce a universal LLM framework named UDDETTS that 1) integrates both ADV and label annotations for the first time, enabling compatibility with diverse types of emotional speech datasets; 2) disentangles complex emotions into the ADV space while addressing sparsity and imbalance issues; 3) provides an interpretable approach for fine-grained emotional TTS control, distinct from traditional label- or description-based prompts. Our work can assist developers in building emotional TTS systems based on large-scale emotional datasets, ultimately enhancing the expressiveness of emotional expression in human-computer interaction.

ETHICS STATEMENT

This research complies with the ICLR Code of Ethics and upholds rigorous standards of academic integrity, legal compliance, and research ethics. All datasets sourced from third-party repositories are used under verified licensing agreements, with explicit documentation provided in Appendix B. Data processing pipelines adhere to privacy-preserving principles and secure storage protocols. For subjective experiments involving human participants, informed consent is obtained and fair compensation is provided. Participant confidentiality is strictly maintained throughout the study. The methodologies and findings reported in this work do not pose significant risks of harm, bias, or misuse. This research is conducted solely for academic purposes, without commercial application, and no conflicts of interest or sponsorship-related influences affect the design, execution, or interpretation of the results.

REPRODUCIBILITY STATEMENT

To ensure the reproducibility of our findings, we take the following measures:

1. **Datasets and baselines.** All datasets and baseline models used in our experiments are listed in Appendix B, C. Fine-tuning strategies and other implementation details for the baselines are also stated in Section 4.3 and 4.5, and all experiments follow the official open-source code and configurations to ensure fairness.

2. **Code availability.** The full implementation of our proposed model, together with configuration files, training scripts, and demos is available at the anonymous repository link: `https://anonymous.4open.science/w/UDDETTS`, ensuring reproducibility and preserving anonymity during the review process.

3. **Experimental details.** Training configurations, evaluation metrics, hardware specifications, and runtime environments are summarized in Section 4.2, with implementation scripts documented in the released code repository.

4. **Theoretical verification.** The algorithmic processes, which require additional explanation, are provided in Appendix E, with step-by-step derivations and explicit clarification of assumptions.

These resources enable independent replication of our experiments and validation of the contributions presented in this paper.

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

## A    THE USE OF LLMS

We confirm that large language models (LLMs), are used exclusively as auxiliary tools for manuscript preparation and refinement. Specifically, LLMs assist in:

1. **Language editing.** Conducting grammar checking, vocabulary optimization, and sentence refinement to enhance clarity and readability.

2. **Visual suggestions.** Providing recommendations for figure preparation, table formatting, and visual coherence to improve presentation quality.

3. **Information retrieval and troubleshooting.** Supporting the search for large-scale English speech datasets, relevant work and literature, and suggesting possible solutions for coding errors.

The design, methodology, original contributions, and code implementation are entirely developed by the human authors. We affirm that all core ideas, theoretical analyses, experimental frameworks, and conclusions reflect human intellectual effort and strictly adhere to academic integrity standards. This statement ensures transparency in AI tool usage while emphasizing the human-led nature of the scientific inquiry.

## B DATASETS

For single-language English, we collect various open-source speech datasets. For general speech datasets, we prioritize samples with human-verified transcriptions to ensure high quality, which helps the model acquire robust TTS capabilities during the first stage training. Due to the high cost of manual annotation, emotional speech datasets are limited in size. We therefore gather diverse types of emotional speech datasets and adapt them to our model using semi-supervised training. Below, we provide a detailed introduction to the datasets used in this paper.

Table 5: Statistics of cleaned speech datasets used in UDDETTS.

| Datasets | #Hours | Type | #Emos | Description |
|---|---|---|---|---|
| MSP (Lotfian & Busso, 2019) | 258.12 | $\mathbb{D}_{S,AL}$ | 8 | Large-scale podcast corpus |
| IEMOCAP (Busso et al., 2008) | 12.28 | $\mathbb{D}_{S,AL}$ | 9 | Acted dialogues in lab |
| CMU-MOSEI (Bagher Z et al., 2018) | 64.23 | $\mathbb{D}_{S,L}$ | 6 | Dialogues from YouTube speakers |
| Expresso (Nguyen et al., 2023) | 1.40 | $\mathbb{D}_{S,L}$ | 13 | Readings and improvisations |
| MELD (Poria et al., 2019) | 8.86 | $\mathbb{D}_{S,L}$ | 7 | TV show dialogues |
| EmoTale (Hjuler et al., 2025) | 0.58 | $\mathbb{D}_{E,AL}$ | 5 | Controlled emotional expressions |
| EU-Emotion (Lassalle et al., 2018) | 11.62 | $\mathbb{D}_{E,AL}$ | 15 | Controlled emotional expressions |
| ESD (Zhou et al., 2022) | 29.07 | $\mathbb{D}_{E,L}$ | 5 | Emotional voice conversion corpus |
| CREMA-D (Cao et al., 2014) | 5.30 | $\mathbb{D}_{E,L}$ | 6 | Controlled emotional expressions |
| EmoV-DB (Adigwe et al., 2018) | 9.48 | $\mathbb{D}_{E,L}$ | 5 | Controlled emotional expressions |
| MEAD (Wang et al., 2020) | 30.12 | $\mathbb{D}_{E,L}$ | 8 | Controlled emotional expressions |
| RAVDESS (Livingstone et al., 2018) | 1.47 | $\mathbb{D}_{E,L}$ | 8 | Controlled emotional expressions |
| Ours | 18.2 | $\mathbb{D}_{S,AL}$ | 6 | Movie dialogues |
| Ours | 83.5 | $\mathbb{D}_{S,L}$ | 9 | Movie dialogues |
| Ours | 1.6 | $\mathbb{D}_{E,AL}$ | 6 | Controlled emotional expressions |
| Ours | 15.3 | $\mathbb{D}_{E,L}$ | 8 | Controlled emotional expressions |
| Total | 551.13 | - | 19 | English emotional speech datasets |
| **Datasets** | **#Hours** | **Type** | **#Emos** | **Description** |
| LibriSpeech (Panayotov et al., 2015) | 987.95 | - | - | Large-scale audiobooks |
| LibriTTS-R (Koizumi et al., 2023) | 578.52 | - | - | Large-scale audiobooks |
| LJSpeech (Ito, 2017) | 23.57 | - | - | Non-fiction books |
| VCTK (Yamagishi et al., 2019) | 43.50 | - | - | Newspaper article readings |
| HiFi-TTS (Bakhturina et al., 2021) | 289.45 | - | - | Large-scale audiobooks |
| HiFiTTS-2 (Langman et al., 2025) | 30000+ | - | - | LibriVox audiobooks |
| Common Voice (Ardila et al., 2020) | 7500+ | - | - | General English Recordings |
| GigaSpeech (Yang et al., 2025) | 10000 | - | - | YouTube, audiobooks, podcasts |
| Total | 49400+ | - | - | English general speech datasets |

## C BASELINES

Here we introduce the ten baselines employed in our experiments. For hyperparameter settings, we follow the official implementations released with the respective papers to reproduce the results. To ensure fairness, all baselines with publicly available pretrained checkpoints and codes are fine-tuned for 10 epochs until convergence solely on our emotional speech datasets, using label prompts as training inputs (e.g., "Angry<|endofprompt|>Content Text"). It is worth noting that, since the training codes for CosyVoice3 and FireRedTTS2 are not publicly available, we do not fine-tune them on our datasets. Instead, we directly perform inference using CosyVoice3-1.5B-RL (plus version api) and FireRedTTS2 checkpoint.

1. **CosyVoice** (Du et al., 2024a) is a scalable multilingual zero-shot TTS model that introduces supervised semantic tokens derived from a speech recognition model. CosyVoice generates semantically aligned speech tokens, enabling improved content consistency and speaker

similarity in synthesized speech. It allows for high-quality, zero-shot voice cloning across multiple languages, while maintaining natural prosody and low-latency synthesis.

2. **CosyVoice2** (Du et al., 2024b) is an advanced TTS model that integrates the LLM with a unified streaming and non-streaming framework. It introduces FSQ for efficient tokens and a chunk-aware causal flow matching model to support diverse synthesis scenarios. These enable it to achieve ultra-low latency synthesis with the first packet latency as low as 150ms, while maintaining high-quality audio output.

3. **CosyVoice3** (Du et al., 2025) is designed for real-world applications, surpassing its predecessor in naturalness, content consistency, speaker similarity, and emotional expressiveness. It introduces a novel speech tokenizer developed by supervised multi-task training, encompassing automatic speech recognition (ASR), language identification (LID), speech emotion recognition (SER), audio event detection (AED), and speaker analysis (SA). It incorporates a differentiable reward model for post-training, enhancing the quality of synthesized speech. It is training data has been expanded from 10,000 hours to 1 million hours.

4. **IndexTTS** (Deng et al., 2025) is an industrial-grade, zero-shot TTS model that enables precise pause control via punctuation marks. while maintaining high-quality audio output. It employs a Conformer-based speech conditional encoder and utilizes BigVGAN2 for speech decoding, achieving high naturalness and speaker similarity. Compared to XTTS, CosyVoice2, F5-TTS, etc., it offers a simpler training process and faster inference speed.

5. **IndexTTS2** (Zhou et al., 2025) is an autoregressive zero-shot TTS model that introduces precise duration control and emotional expressiveness. It supports two generation modes: one that explicitly specifies token counts for accurate duration, and another that generates speech freely while preserving prosody. The model decouples timbre and emotion, enabling independent control over both aspects. Additionally, it incorporates GPT latent representations and a three-stage training paradigm to enhance speech clarity. IndexTTS2 outperforms existing models in word error rate, speaker similarity, and emotional fidelity.

6. **FireRedTTS** (Guo et al., 2025) comprises three main components: a data processing pipeline that transforms massive raw audio into high-quality TTS datasets with rich annotations; a LLM-based TTS model that compresses speech signals into discrete semantic tokens via a semantic-aware speech tokenizer; and a two-stage waveform generator that decodes the semantic tokens into waveforms. FireRedTTS demonstrates solid in-context learning capabilities, achieving zero-shot voice cloning and few-shot adaptation.

7. **FireRedTTS2** (Xie et al., 2025) is a long-form streaming TTS model developed for multi-speaker dialogue generation, addressing limitations in existing models regarding stability, speaker switching, and prosody coherence. It introduces a 12.5Hz streaming speech tokenizer that accelerates inference, extends maximum dialogue length.

8. **Spark-TTS** (Wang et al., 2025) leverages the LLM for high-quality TTS. It employs Bi-Codec, a single-stream speech codec that decomposes speech into two complementary token types: low-bitrate semantic tokens for linguistic content and fixed-length global tokens for speaker-specific attributes. It allows for controllable speech generation through adjustable parameters such as gender, pitch, and speaking rate.

9. **F5-TTS** (Chen et al., 2025c) utilizes flow matching with a Diffusion Transformer (DiT) backbone. It pads the input text with filler tokens to match the length of the target speech. It integrates ConvNeXt for refining text representations and introduces an inference-time Sway Sampling strategy, which improves model efficiency and output quality.

10. **VALL-E** (Chen et al., 2025b) is the first neural codec language model developed by Microsoft for zero-shot TTS. It utilizes discrete tokens derived from a neural codec model and frames TTS as a conditional language modeling task. It can synthesize high-quality personalized speech from a 3-second acoustic prompt.

## D    LABEL STATISTICS

We collect emotion label statistics in all datasets and map them to individual label tokens. Table 6 shows the sample count for each label, and Figure 6 shows the distribution of some emotion samples in the ADV space.

Table 6: Emotion labels, corresponding label tokens, and sample counts used in UDDETTS.

| Token | Emotion(s) | Samples | Token | Emotion(s) | Samples |
|---|---|---|---|---|---|
| 0 | Unknown | 42235 | 5 | Fearful | 6654 |
| 1 | Sad, Frustrated, Hurt | 27135 | 6 | Sleepiness, Bored | 4331 |
| 2 | Angry | 35258 | 7 | Neutral, Narration | 68042 |
| 3 | Confused, Worried | 7149 | 8 | Surprise, Excited | 10214 |
| 4 | Disgust, Contempt | 14972 | 9 | Happy, Amused, Laughing | 57433 |

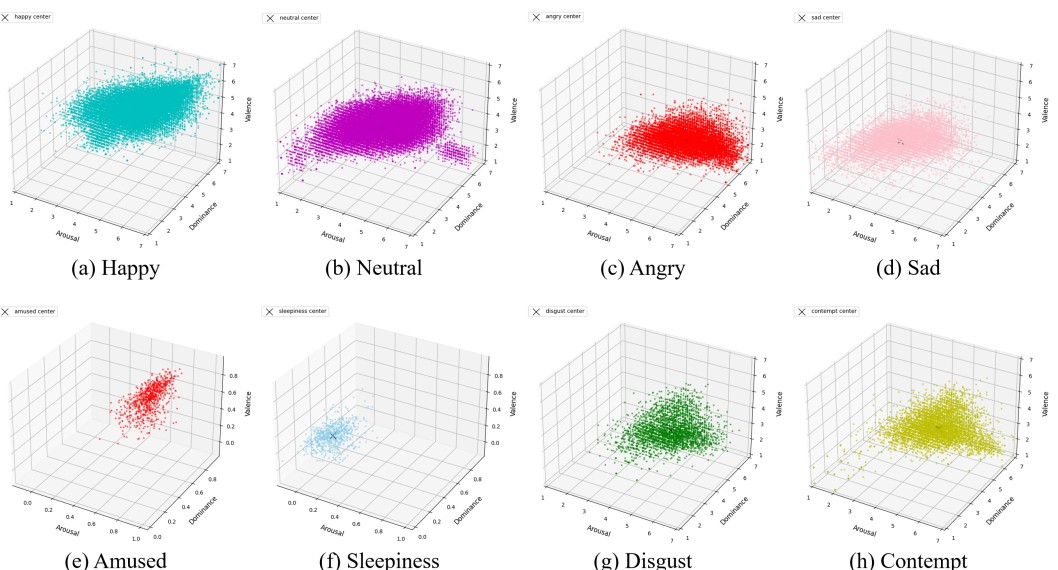

(a) Happy     (b) Neutral     (c) Angry     (d) Sad

(e) Amused     (f) Sleepiness     (g) Disgust     (h) Contempt

Figure 6: The distribution of some emotional samples in the ADV space. Each emotion tends to form a distinct cluster.

# E  ADV STATISTICS AND NONLINEAR BINNING ALGORITHM

We perform distribution statistics of the ADV values across all $\mathbb{D}_{S,AL}$ and $\mathbb{D}_{E,AL}$ datasets. The nonlinear binning algorithm is then applied along the three dimensions, and the resulting binning scheme is illustrated in Figure 7. The detailed clustering-based nonlinear binning procedure of the ADV quantizer is provided in Table 7.

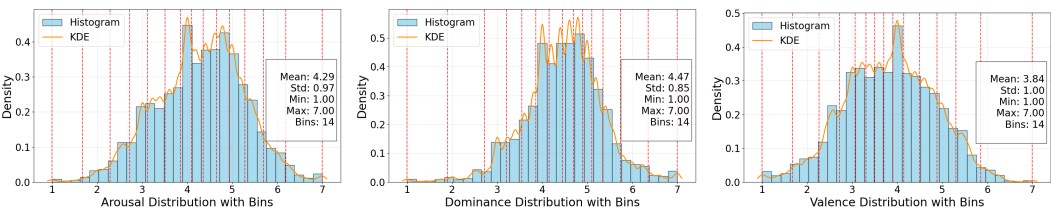

Figure 7: The histograms and kernel density estimations of all training samples along the three dimensions of the ADV space are shown, with the x-axis representing the continuous ADV values. Red dashed lines indicate the division of each dimension into 14 bins.

Table 7: Clustering-based nonlinear binning algorithm for the ADV space.

| Step | Description & Formula |
|------|----------------------|
| **Mapping** | Merged dataset $\mathbb{D}_{S|E,AL} = \{\boldsymbol{x}_i\}_{i=1}^N$, $\boldsymbol{x}_i = (a_i, d_i, v_i) \in \mathbb{R}^3$. Linear map $f : [\min_c, \max_c] \to [1,7]$: $\boldsymbol{x}_{c,i} = f(\boldsymbol{x}_{c,i})$, $c \in \{a,d,v\}$. |
| **#Clusters $K$** | **Step 1:** $N = |\mathbb{D}_{S|E,AL}|$, the maximum $K_{\max} \le \lfloor \sqrt[3]{N} \rfloor$ to probe. Initialize hash-map $\mathcal{H} : k \mapsto (k, \bar{s}_k, \widehat{\sigma}_k, R_k)$. **Step 2:** For $k = 2$ to $K_{\max}$ with step $s$: run k-means $R$ times, compute silhouette score $s_k^{(r)}$, $\bar{s}_k = \frac{1}{R}\sum_{r=1}^R s_k^{(r)}$, and $\hat{\sigma}_k$, store $(k, \bar{s}_k, \hat{\sigma}_k, R)$ in $\mathcal{H}$. **Step 3:** Sort $\mathcal{H}$ by decreasing $\bar{s}_k$. For top $M = \lceil |\mathcal{H}|/4 \rceil$ candidates $\mathbb{C} = \{k_1, k_2, \ldots, k_M\}$. For each $k \in \mathbb{C}$, refine by evaluating neighbors $k-1$ and $k+1$, and insert into $\mathcal{H}$. Report: $$K = \underset{k \in keys(\mathcal{H})}{\arg\max}\, \overline{(s_k - \lambda\widehat{\sigma}_k)}.$$ |
| **Clustering** | Run k-means in $\mathbb{R}^3$ with selected $K$, obtain clusters $\boldsymbol{C}_1, \ldots, \boldsymbol{C}_K$ and centroids $\{\boldsymbol{\mu}_j\}_{j=1}^K$, $\boldsymbol{\mu}_j = (\mu_{j,a}, \mu_{j,d}, \mu_{j,v})$. Objective: $$\min_{C_1,\ldots,C_K} J = \sum_{j=1}^K \sum_{x_i \in C_j} ||x_i - \mu_j||^2,$$ |
| **Boundaries** | For each axis $c \in \{a,d,v\}$ take the set of center coordinates: $\mathbb{M}_c = \{\mu_{1,c}, \mu_{2,c}, \ldots, \mu_{K,c}\}$, sort $\mathbb{M}_c$: $m_{c,(1)} \le m_{c,(2)} \le \cdots \le m_{c,(K)}$. Midpoint Boundaries: $t_{c,i}^{\mathrm{mid}} = \frac{1}{2}(m_{c,(i)} + m_{c,(i+1)})$; weighted Boundaries: $$t_{c,i}^{\mathrm{w}} = \frac{|\boldsymbol{C}_{\pi_c(i)}|\, m_{c,(i)} + |\boldsymbol{C}_{\pi_c(i+1)}|\, m_{c,(i+1)}}{|\boldsymbol{C}_{\pi_c(i)}| + |\boldsymbol{C}_{\pi_c(i+1)}|}, i = 1, \ldots, K-1.$$ $n_i = |\boldsymbol{C}_i|$, $\sigma_i^2 = \mathrm{Var}(x_c; x \in \boldsymbol{C}_i)$, $r_i = max(\sigma_i^2, \sigma_{i+1}^2)/min(\sigma_i^2, \sigma_{i+1}^2)$, $t_{c,i} = t_{c,i}^{\mathrm{mid}} + 1_{\{r_i > 2\}}(t_{c,i}^{\mathrm{w}} - t_{c,i}^{\mathrm{mid}})$. |
| **Tokens** | Given bins $\{t_{c,i}\}$, map $x_c$ to tokens by: $$\tau_c = 1 + \sum_{i=1}^{K-1} 1_{\{x_c > t_{c,i}\}}, c \in \{a,d,v\}.$$ |

## F  THE TEST SET

Table 8: Some examples of test text corpus with emotional content.

| Emotion | Text |
|---------|------|
| Neutral | For the twentieth time that evening the two men shook hands. |
| Neutral | She open the door and walk into the room. |
| Neutral | The meeting start promptly at nine in the morning. |
| Happy | I'm so happy to be friends with you. |
| Angry | I'm very angry now because you did not arrive on time! |
| Sad | Lost wallet, missed last bus, tears drown my voiceless night. |
| Sleepiness | I'm tired because I had to work overtime until evening. |
| Mixed | I love you so much, I can't live without you! |

We construct a test text corpus comprising two standard test sets, LibriSpeech-test-clean[4] and SeedTTS-test-en[5], which are used for evaluating objective metrics such as WER, SS, and ES, STOI

---

[4]https://www.openslr.org/12/

[5]https://github.com/BytedanceSpeech/seed-tts-eval

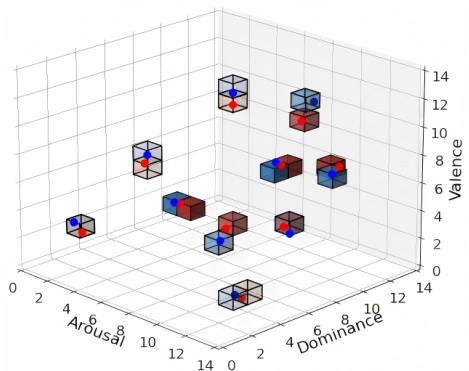

Figure 8: Text-derived (blue) and speech-derived (red) ADV values within their control units for ten emotionally-biased sentences.

and PESQ-WB. For subjective evaluation, we design a separate corpus comprising 20 neutral sentences for controllable synthesis and 10 emotionally-biased sentences for end-to-end emotional TTS. The neutral texts are randomly sampled and filtered using the Senta model[6], retaining only those with over 90% confidence as neutral. The emotionally-biased sentences are generated by GPT-5 and manually selected by three evaluators from 50 candidates. These texts are semantically unambiguous and contain inherent emotional cues, avoiding interpretive ambiguity. All texts are unseen during training, eliminating overfitting concerns. Examples from the corpus are shown in Table 8.

# G   SAM SYSTEM

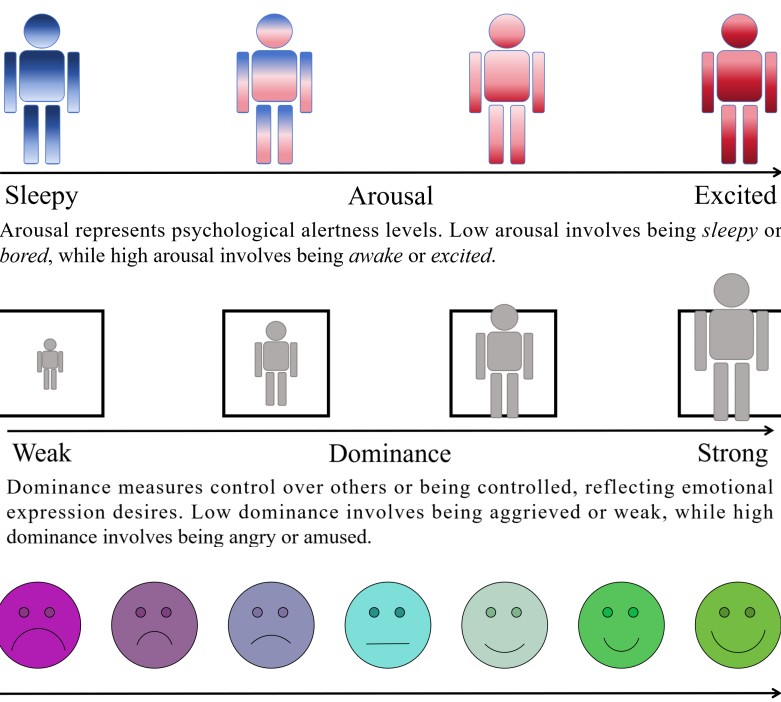

Arousal represents psychological alertness levels. Low arousal involves being *sleepy* or *bored*, while high arousal involves being *awake* or *excited*.

Dominance measures control over others or being controlled, reflecting emotional expression desires. Low dominance involves being aggrieved or weak, while high dominance involves being angry or amused.

Valence (also known as pleasure) represents the emotional positivity or negativity, such as being *sad* or *angry* as low valence, while being *happy* or *excited* as high valence.

Figure 9: Visualization of the three ADV dimensions using the SAM system.

---

[6]https://github.com/baidu/Senta

Inspired by Morris (1995), we use the Self-Assessment Manikin (SAM) system to visually and intuitively manipulate $x_{adv}$, enabling fine-grained control and helping evaluators intuitively understand the decoupled emotional dimensions for accurate ranking. Each ADV dimension is represented by a graphic character arrayed along a continuous scale, as shown in Figure 9.

# H  ROBUSTNESS ANALYSIS

To further validate the robustness of UDDETTS under control, we conduct evaluations from the following perspectives:

1. **Label robustness under varying data resources.** We examine whether labels with sparse training samples can still be controlled effectively. In the first experiment, we select five emotions with stepwise decreasing sample sizes to test the model's performance under both high-resource and low-resource conditions. As shown in Table 1, UDDETTS achieves more accurate overall emotional expression compared with baselines. Table 9 further details the results across the five emotions, demonstrating that UDDETTS performs particularly well on low-resource categories.

2. **Robustness to unseen emotion labels.** For emotions absent in the training set, we assess whether the synthesized speech aligns with the label using an emotion confusion matrix. Table 9 reports results for two such labels.

3. **Robustness to unseen ADV regions.** Although the nonlinear binning algorithm and semi-supervised training expand the soft coverage of the ADV space (regions close to training samples), certain hard unseen regions (far from all training distributions) remain challenging for high-quality synthesis. Table 10 presents MOS and UTMOS results in some of these unseen ADV regions.

4. **ADV-label conflict robustness test.** For mixed emotions in overlapping cluster regions, a single ADV value may correspond to multiple potential emotion labels. We test this by controlling label tokens (angry, sad, happy, neutral) while fixing the ADV value in angry-sad overlapping regions. Results show minimal perceptual differences between angry, sad, and neutral labels. With the happy token, speech retains the angry-sad style but exhibits higher pitch and sporadic laughter, revealing inherent conflict between this ADV value and the happy label. It is noteworthy that the autoregressively predicted labels from ADV inputs remain within emotionally consistent categories, confirming the dominant role of ADV in emotion control.

Table 9: Robustness test results of five labels and some unseen labels.

| Acc. \ Emotions  Models | Neutral | Happy | Angry | Disgust | Sleepiness | loving | anxious |
|---|---|---|---|---|---|---|---|
| UDDETTS | 1.000 | 1.000 | 0.975 | **0.840** | **0.890** | **0.775** | **0.605** |
| CosyVoice | 1.000 | 0.975 | 0.900 | 0.635 | 0.695 | 0.375 | 0.310 |
| CosyVoice2 | 1.000 | 1.000 | 0.975 | 0.650 | 0.700 | 0.405 | 0.330 |
| CosyVoice3 | 1.000 | 1.000 | **1.000** | 0.795 | 0.790 | 0.620 | 0.550 |
| IndexTTS | 1.000 | 1.000 | 0.910 | 0.675 | 0.705 | 0.320 | 0.545 |
| IndexTTS2 | 1.000 | 1.000 | 0.945 | 0.770 | 0.795 | 0.410 | 0.580 |
| FireRedTTS | 1.000 | 0.985 | 0.875 | 0.665 | 0.720 | 0.375 | 0.315 |
| FireRedTTS2 | 1.000 | 0.780 | 0.880 | 0.670 | 0.725 | 0.560 | 0.565 |
| Spark-TTS | 1.000 | 1.000 | 0.950 | 0.805 | 0.855 | 0.600 | 0.520 |
| F5-TTS | 1.000 | 1.000 | 1.000 | 0.785 | 0.875 | 0.575 | 0.495 |
| VALL-E | 1.000 | 0.975 | 0.810 | 0.450 | 0.570 | 0.250 | 0.300 |

Table 10: Evaluation on unseen soft and hard ADV values

| UDDETTS | Soft | | | Hard | | |
|---|---|---|---|---|---|---|
| | [14,1,1] | [6,1,1] | [3,4,10] | [1,7,14] | [1,14,14] | [1,14,7] |
| MOS | 4.30 | 4.10 | 4.08 | 3.65 | 3.56 | 3.60 |
| UTMOS | 4.20 | 3.98 | 4.15 | 3.85 | 3.20 | 3.43 |

# I   IMPACT OF ADV CONTROL ON PROSODIC FEATURES

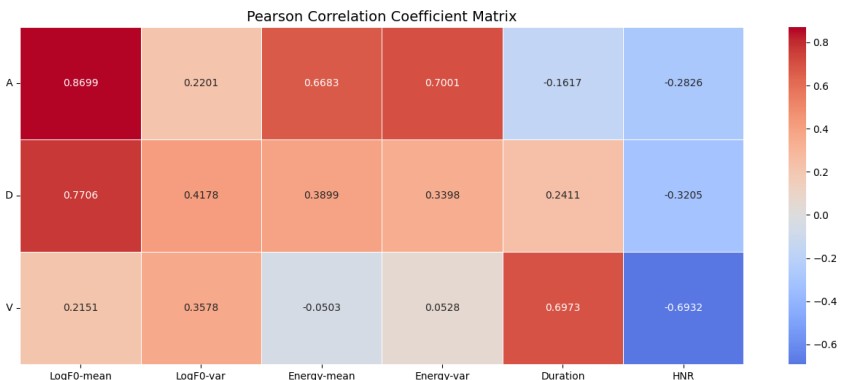

Figure 10: The Pearson correlation coefficient matrix showing the relationship between each ADV dimension and prosodic statistics.

To study the impact of ADV control on emotional representations, we vary all values of $x_{adv} \in \mathbb{Z}^3_{[1,14]}$ to synthesize emotional speech and extract their prosodic features, including the mean and variance of $log$ F0 and energy, as well as duration and harmonic-to-noise ratio (HNR). We compute the Pearson correlation between each ADV dimension and these prosodic statistics. The results in Figure 10 show that Arousal and Dominance are significantly correlated with $log$ F0 and energy, indicating their role in controlling the excitement and intensity of emotion. Valence is correlated with HNR, which reflects voice quality variations linked to emotional changes (Borchert & Dusterhoft, 2005), and it also affects the shape of the mel-spectrogram in Figure 11, indicating its influence on emotional polarity. Its correlation with duration is likely due to laughter in high-valence speech. To further analyze the variation of emotional speech along the ADV axes, Table 11 reports the changes in prosodic features when slightly perturbing the ADV values around eight emotion cluster centers. Specifically, we adjust each dimension of ADV by $\pm 4$ (denoted as "+" for upward shift and "–" for downward shift), and measure the corresponding changes in average $log$ F0, energy, duration, and HNR. We observe that positive arousal is associated with higher pitch and energy. Similarly, positive dominance not only increases pitch and energy but also narrows their variation ranges, and it is further associated with longer durations. In contrast, valence has little effect on pitch and energy but tends to reduce HNR variations, influencing emotional polarity. Overall, the results align with the intrinsic characteristics of each ADV dimension, supporting the effectiveness of our approach in capturing and interpreting emotional variations in speech.

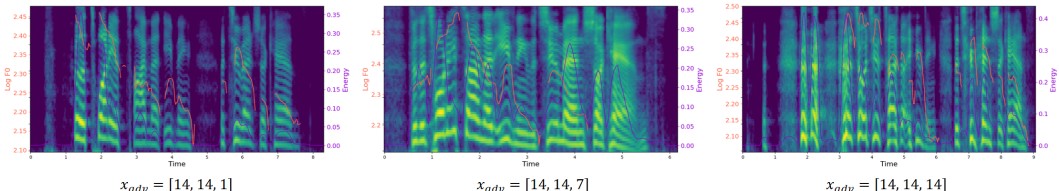

Figure 11: The patterns of F0 contours observed in the mel-spectrogram vary as a function of valence.

Table 11: Comparisons of F0, energy, duration, and HNR for eight emotions across different ADV patterns.

| Emotions | Patterns | F0 (mean) | Energy (mean) | Duration (mean) | HNR |
|---|---|---|---|---|---|
| Happy | +A +D +V | +8.0 | +0.038 | +1.2 | -2.4 |
| | -A +D +V | -2.6 | +0.028 | +0.4 | -2.3 |
| | +A -D +V | +6.7 | +0.003 | +0.6 | -2.3 |
| | +A +D -V | +7.9 | +0.031 | -1.2 | +1.7 |
| | -A -D +V | -7.6 | -0.032 | +0.3 | -2.0 |
| | -A -D -V | -7.8 | -0.040 | -2.0 | +1.9 |
| Angry | +A +D +V | +6.5 | +0.040 | -0.1 | -2.0 |
| | -A +D +V | -2.4 | +0.032 | +0.4 | -1.8 |
| | +A -D +V | +5.2 | -0.015 | -0.3 | -1.7 |
| | +A +D -V | +6.0 | +0.045 | -0.5 | +1.5 |
| | -A -D +V | -6.4 | -0.033 | +0.6 | -1.7 |
| | -A -D -V | -6.7 | -0.036 | -0.1 | +1.6 |
| Sad | +A +D +V | +5.8 | +0.033 | +0.2 | -2.3 |
| | -A +D +V | +1.8 | -0.012 | +0.3 | -1.4 |
| | +A -D +V | +3.4 | +0.028 | -0.2 | -1.5 |
| | +A +D -V | +5.1 | +0.043 | -0.4 | +1.5 |
| | -A -D +V | -4.9 | -0.028 | +0.3 | -0.9 |
| | -A -D -V | -5.2 | -0.024 | -0.3 | +2.2 |
| Disgust | +A +D +V | +4.8 | +0.023 | +0.2 | -1.7 |
| | -A +D +V | -0.9 | -0.012 | +0.1 | -0.9 |
| | +A -D +V | +3.4 | -0.005 | -0.0 | -0.4 |
| | +A +D -V | +4.6 | +0.026 | -0.4 | +0.4 |
| | -A -D +V | -5.0 | -0.026 | -0.2 | -0.1 |
| | -A -D -V | -5.1 | -0.023 | -0.3 | +1.2 |
| Surprise | +A +D +V | +5.2 | +0.045 | +0.8 | -2.1 |
| | -A +D +V | -3.4 | +0.010 | +0.5 | -1.8 |
| | +A -D +V | +4.7 | +0.007 | +0.2 | -1.7 |
| | +A +D -V | +5.0 | +0.040 | -0.1 | +1.5 |
| | -A -D +V | -5.3 | -0.039 | -0.3 | -1.0 |
| | -A -D -V | -5.6 | -0.042 | -0.3 | +2.3 |
| Fearful | +A +D +V | +3.5 | +0.031 | +0.3 | -1.0 |
| | -A +D +V | -1.8 | -0.025 | +0.1 | -0.3 |
| | +A -D +V | -0.6 | -0.005 | -0.1 | -0.1 |
| | +A +D -V | +2.5 | +0.034 | -0.3 | +0.2 |
| | -A -D +V | -3.4 | -0.034 | +0.1 | +0.2 |
| | -A -D -V | -3.8 | -0.032 | -0.2 | +0.5 |
| Confused | +A +D +V | +5.2 | +0.040 | -0.1 | -1.8 |
| | -A +D +V | -3.8 | -0.020 | +0.3 | -1.2 |
| | +A -D +V | +4.2 | +0.003 | -0.2 | -1.3 |
| | +A +D -V | +4.9 | +0.005 | -0.4 | +1.2 |
| | -A -D +V | -5.7 | -0.029 | +0.1 | -0.9 |
| | -A -D -V | -5.4 | -0.030 | -0.3 | +1.5 |
| Sleepiness | +A +D +V | +2.1 | +0.010 | +0.0 | -2.9 |
| | -A +D +V | -0.9 | -0.007 | +0.2 | -2.2 |
| | +A -D +V | +1.1 | +0.002 | -0.1 | -2.0 |
| | +A +D -V | +2.2 | +0.010 | +0.1 | +0.4 |
| | -A -D +V | -2.4 | -0.013 | -0.1 | -1.5 |
| | -A -D -V | -2.6 | -0.012 | -0.2 | +1.8 |

