## Revision Summary

To facilitate the review process, we summarize the supplementary content added to the paper. In response to the reviewers' comments, we have included the following additional experiments and analyses:

1. Added comparison with CosyVoice + ADV in Section 4.3 (Lines 373-402) and Table 1.

2. Included sensitivity analysis on the number of bins in Section 4.4 (Lines 424-429) and Figure 5.

3. Expanded comparison between UDDETTS and the traditional non-LLM-based Emo-Sphere++ model in Section 4.4 (Lines 465-476), Table 3, and the rightmost subfigure of Figure 4.

4. Supplementary results on the ADV predictor, multi-task speech tokenizer, and emotional consistency between text semantics and synthesized speech in Section 4.5 (Lines 495-501) and Figure 8 in Appendix F.

5. Further explanation of the semi-supervised learning ablation in Section 4.6 (Lines 515-519).

6. Additional analysis of ADV-label conflict robustness in Appendix H.