# OpenReview forum: "UDDETTS: Unifying Discrete and Dimensional Emotions for Controllable Emotional Text-to-Speech"
_ICLR.cc/2026/Conference — Submitted to ICLR 2026_

### Official Review · Reviewer_2DR5 · 2025-10-27

**Soundness:** 2
**Presentation:** 4
**Contribution:** 3
**Rating:** 6
**Confidence:** 3

**Summary:**

This paper proposes a controllable emotional TTS framework that unifies discrete emotion labels with a dimensional Arousal–Dominance–Valence (ADV) space. Contributions can be summarized as:
- a semi-supervised neural-codec LLM that can take either labels or quantized ADV tokens as control input;
- a nonlinear binning scheme that discretizes the ADV space into controllable units;
- an ADV predictor that infers pseudo-ADV from text for end-to-end, text-only emotional synthesis.

**Strengths:**

- Unifying discrete and dimensional emotion control in LLM-based TTS is well-motivated and really novel for fine-grained affect control.
- This paper presents solid and extensive experiments.
- This paper provides code at an anonymous link.

**Weaknesses:**

- ADV ranges are normalized to [1,7], and bins are chosen via a CLT-inspired heuristic plus clustering. It’s unclear how sensitive control linearity is to the chosen number of bins and cluster variability.
- Although the paper aims to achieve fine-grained and interpretable emotional control through the continuous (ADV) space, the amount of training data with ground-truth ADV annotations appears to be very limited. Most emotional datasets only provide discrete emotion labels, while ADV values are available for a small subset. As a result, the ADV predictor and the overall controllability of the system rely heavily on pseudo-ADV values inferred from semi-supervised learning rather than real annotated data. This raises concerns about the precision and reliability of the ADV mapping, especially for subtle or compound emotions. The scarcity of reliable ADV-labeled samples might constrain the model’s ability to learn accurate continuous emotion representations, which somewhat contradicts the paper’s goal of achieving fine-grained control in the ADV space.
- The system employs both an ADV predictor and a label predictor. However, the paper does not clearly explain how these two emotion sources interact or which one dominates when their predictions disagree. Since the final emotional output depends on the fusion of both, inconsistencies between the predicted ADV vectors and categorical labels could lead to unstable or conflicting emotional expressions. Moreover, no quantitative analysis (e.g., disagreement rate, calibration curve, or preference correlation) is provided to demonstrate whether the two predictors are aligned. This ambiguity raises concerns about the reliability and interpretability of the emotional control, which is central to the paper’s claimed contribution.

**Questions:**

- How robust are the nonlinear bin boundaries across different training splits?
- How often does the ADV predictor disagree with the LLM-predicted label, and which path dominates the final emotion?

---

> ### Author Response · Authors · 2025-11-21
> **Re weaknesses and questions**
>
> ## Re weakness 1 and question 1: Sensitivity Analysis of Bin Count m
>
> This sensitivity analysis has been included in Section 4.4 (Lines 424-429) and Figure 5 to demonstrate the robustness of our nonlinear binning method.
>
> - The bin count $m=14$ is determined algorithmically as the optimal value. Our sensitivity analysis confirms that the system remains robust within the range $m∈[12, 16]$, maintaining relatively linear emotional control across the ADV space.
>
> - Experimental Validation: We used emotion2vec [[1]](https://github.com/ddlBoJack/emotion2vec) to extract emotional features and t-SNE visualization to analyze sample distributions. Results confirm that excessive m-values cause certain intermediate ADV-controlled emotions to deviate from the expected regression curve.
>
> - When $m<12$: Emotional transitions become rapid and non-smooth at extreme ADV values (both low and high ranges), though central regions remain stable.
>
> - When $m>14$: Insufficient samples per control unit lead to degraded generalization, unnatural transitions at soft ADV values, and occasional emotional discontinuities.
>
>
> ## Re weakness 2, 3 and question 2: Data Balance & ADV-Label Alignment
>
> - Data Proportion:
>
>   ADV-annotated data comprises 300+ hours (54.87%) of our emotional speech datasets. We carefully regulated training data ratios to prevent annotation scarcity, with further mitigation through semi-supervised learning and nonlinear binning (Figure 4).
>
> - Credibility of Pseudo-ADV Values and ADV Predictor Performance:
>   - Pseudo-ADV values are exclusively utilized in the end-to-end emotional TTS task. To validate the emotional consistency between text semantics (represented by pseudo-ADV values) and synthesized speech, we have supplemented the following experimental evidence in Section 4.5 (Lines 495-501) and Figure 8 (Appendix F).
>   Figure 8 intuitively proves the accuracy and reliability of ADV mapping.
>
>   - The ADV predictor achieves a final RMSE of 1.25, slightly higher than the RMSE of 0.68 achieved by the multi-task speech tokenizer on the SADVR task, which predicts ADV directly from speech. The difference arises because some texts are semantically ambiguous, leading to weaker alignment between the textual semantic emotion and the vocal emotional expression. However, in practice, this level of deviation only causes subtle perceptual difference in emotional control. We invite you to verify this by listening to the synthesized speech for adjacent ADV values on our Demo page, where the emotional progression remains smooth.
>
> - ADV-Label Relationship:
>
>     In Appendix H, we supplement the ADV-label conflict robustness analysis. The prosodic metrics in Table 11 further elucidate these ADV-label relationships.
>     - When an ADV value is specified, the corresponding emotion label remains relatively clear. As shown in Figure 6, numerous ADV values form emotional clusters that predominantly map to the same discrete label. In most cases, ADV and labels maintain a many-to-one mapping without conflict—you can consider ADV values as detailed emotional attributes that refine the coarse-grained labels. For blended emotions like "loving" located in overlapping cluster regions, certain ADV values might correspond to multiple potential labels.
>     - Tests in angry-sad overlapping regions:
>         - Minimal perceptual differences between angry, sad and neutral labels, confirming ADV's dominant role.
>         - When the happy token, speech retains the angry-sad style but exhibits higher pitch and sporadic laughter, revealing inherent conflict between this ADV value and the happy label.
>         - During inference, UDDETTS primarily negotiates between similar emotions in overlapping regions while maintaining ADV's dominant control.
>
> [1] https://github.com/ddlBoJack/emotion2vec

---

> > ### Comment · Reviewer_2DR5 · 2025-11-21
> >
> > Thank you for your further explanation. I respectfully offer some suggestions that are not directly related to the content:
> >
> > - ICLR allows unlimited modifications to the PDF. I recommend that the author highlight the modified sections in blue and provide an overall comment listing the specific changes made to the PDF.
> >
> > - I suggest that the author use markdown format with different heading levels and bold text to improve the readability of the responses. Additionally, it would be helpful to reference specific chapters, tables, or figures from the modified PDF in the response.

---

> > > ### Author Response · Authors · 2025-11-25
> > >
> > > Thank you very much for your suggestion. We have made our best efforts to address all the reviewers' questions and have supplemented the manuscript with relevant experiments. We also hope that you will pay attention to our other comments and replies for a better understanding of our work. All experimental results have been incorporated into the revised manuscript, which has been updated to meet the 10-page limit in accordance with ICLR requirements. We summarize the supplementary content added to the paper:
> > >
> > >
> > > - Added comparison with CosyVoice + ADV in Section 4.3 (Lines 373-402) and Table 1.
> > >
> > > - Included sensitivity analysis on the number of bins in Section 4.4 (Lines 424-429) and Figure 5.
> > >
> > > - Expanded comparison between UDDETTS and the traditional non-LLM-based EmoSphere++ model in Section 4.4 (Lines 465-476), Table 3, and the rightmost subfigure of Figure 4.
> > >
> > > - Supplementary results on the ADV predictor, multi-task speech tokenizer, and emotional consistency between text semantics and synthesized speech in Section 4.5 (Lines 495-501) and Figure 8 in Appendix F.
> > >
> > > - Further explanation of the semi-supervised learning ablation in Section 4.6 (Lines 515-519).
> > >
> > > - Additional analysis of ADV-label conflict robustness in Appendix H.

---

### Official Review · Reviewer_8NAV · 2025-10-31

**Soundness:** 2
**Presentation:** 1
**Contribution:** 2
**Rating:** 2
**Confidence:** 5

**Summary:**

This paper proposes UDDTTS, an LLM-based TTS using an ADV space to model emotional representations for expressive speech synthesis. While the idea is interesting, the paper lacks methodological clarity, strong experimental validation, and a comprehensive review of related LLM-based TTS work. The results also do not show clear advantages over existing methods.

**Strengths:**

The work introduces a potentially useful direction for controllable emotional TTS by modeling ADV in LLMs.

**Weaknesses:**

Your proposed UDDTTS does not outperform other approaches in terms of MOS, UTMOS, WER, SS, and STOI. I suggest further improving these metrics through more refined method design.

Methodology is not well written. Please define the symbols before using them. I am confused with the method design.

The generated speech quality is not good with unclear pronunciations, which is not common in the existing TTS models. I am wondering if including ADV is the reason why the speech intelligence is getting worse. I would suggest improving the performance further with more advanced techniques.

The literature review on LLM-based TTS approaches is relatively limited, and a more comprehensive investigation is recommended.

I am not fully convinced by how you disentangle complex emotions in the ADV space while addressing sparsity and imbalance issues. Could you provide experimental evidence to support this claim?

It is unclear why AB preference tests were not included, as they are commonly used to assess perceptual differences in TTS quality.

**Questions:**

How did you prove that you capture the continuity of emotion distributions?

Having only 12 listeners for the subjective evaluation is insufficient for a comprehensive assessment of TTS models. For each listener, how many samples were evaluated? Were these samples randomly selected from the test set or manually chosen?

How did you create and process spontaneous emotion datasets and elicited emotion datasets?

What does Z1 mean?

Why do you assume that Xspk can effectively represent the speaker embedding while excluding emotional representations? Could you elaborate on this assumption and provide evidence or verification?

How do you demonstrate that your proposed speech tokenizer captures rich emotional information? What are the key differences between your tokenizer and CosyVoice’s speech tokenizer?

I am also unclear about the motivation, design, and working mechanism of the ADV predictor. Could you explain this in more detail? Is the speech signal considered in its process, or is it modeled solely based on textual emotion inputs? If speech is not incorporated, the emotional states between speech and text might differ. How did you address this issue?

Will you release your testests?

---

> ### Author Response · Authors · 2025-11-14
> **Re weaknesses 1, 2, 3, 4, 6**
>
> ## Re weakness 1:
>
> - UDDETTS achieves comparable scores to state-of-the-art LLM-based TTS models (e.g., Cosyvoice3, IndexTTS2) in naturalness, WER, speaker similarity, and STOI, demonstrating strong synthesis capability and robustness trained on 49k+ hours.
> - However, this section focuses on label-controlled emotional TTS, where improvements in emotional metrics are the focus. Table 1 shows significant improvements in emotion-related metrics: Pm/Rm of the emotion confusion matrix, emotion similarity, and PESQ-WB. These confirm UDDETTS's higher accuracy and expressiveness in label-controlled emotional TTS.
>
> Note: stronger emotional expression may slightly reduce WER and SS due to less precise articulation in expressive speech.
>
> ## Re weakness 2:
> - We have uploaded the new version of the paper and put the revision summary of the paper in the attached materials.
> - We have strictly followed ICLR notation guidelines. All symbols in Sec 3.1.1 formulas and Figure 2 are explicitly defined, with losses contextualized and Table 6 (Appendix) containing step-by-step annotations. Could you specify which symbols need clarification?
>
> ## Re weakness 3:
> - Our demo page provides generated speech samples from all compared models, while none of the baseline models incorporate the ADV. UDDETTS demonstrates no significant pronunciation issues according to both multi-evaluator subjective assessments and objective metrics. In fact, it produces clearer articulation than several other LLM-based models, as reflected in its lower WER (below 3% already indicates clear and accurate pronunciation).
> - Your subjective differences may stem from our second-stage training dataset, which comprises emotional English speech from diverse nationalities, genders, and age groups. This diversity naturally introduces some connected speech patterns compared to standard slow-paced pronunciation, but this is unrelated to the ADV.
>
> ## Re weakness 4:
> Our introduction (Lines 046-063) already cites major LLM-based TTS models (e.g., CosyVoice1-3, IndexTTS1-2, FireRedTTS1-2, F5-TTS, Seed-TTS, Spark-TTS, VALL-E), with comprehensive comparisons in experiments. The related work section (Lines 117-141) further categorizes existing approaches, but we prioritized recent (past three years) publications to conserve space for methodological contributions. Appendix C (Lines 916-966) provides detailed coverage of these state-of-the-art LLM-based TTS models.
>
> ## Re weakness 6:
> - AB Preference Tests Already Included: We confirm that AB preference tests with p-values were already included in Table 4 of the initial submission. We are unclear why the reviewer stated they were absent. We hope the reviewer can examine our work more carefully!
> - Additional ABX Tests Added: Furthermore, we have supplemented the comparison by including ABX tests between UDDETTS and EmoSphere++ in Table 3.
> - MOS and UTMOS are widely accepted methodologies for assessing TTS quality.

---

> ### Author Response · Authors · 2025-11-14
> **Re questions 1-5**
>
> ## Re question 1:
>
> - The three-dimensional floating-point ADV values inherently capture continuous emotional distributions, as visualized in Figures 6.
> - Our nonlinear binning algorithm (Table 7) enables fine-grained control while preserving emotional continuity
> - Table 2, Figure 4 and Figure 5 demonstrate that UDDETTS achieves fine-grained, interpretable, and linear emotion control along three psychological dimensions.
> - Demo page provides multiple examples where clear emotional progression can be perceived
>
> ## Re question 2:
> - Complete test set details are provided in Appendix F.
> - Subjective Evaluation involved 12 professional listeners meeting standard MOS requirements
> - Each compensated participant evaluated 4380 samples across three test categories:
>     - 20 neutral texts × 11 emotion labels × 12 models
>     - 20 neutral texts × 14 granularity levels × 3 dimensions × 2 models
>     - 20 texts × 3 models for ABX preference tests
>
> - Text selection as shown in Lines 1906-1103:
>     - 20 neutral texts: randomly sampled from LibriSpeech-test-clean and SeedTTS-test-en, filtered via Senta model (≥90% neutral confidence).
>     - 10 emotional texts: GPT-5 generated, manually selected, semantically unambiguous.
>
> ## Re question 3:
>
> - Most English emotional speech datasets are open-source (Appendix B)
> - Our proprietary datasets follow the same creation methodology, utilizing multi-annotator labeling and professional voice recordings.
> - The distinction between spontaneous and elicited emotion datasets is clarified in Lines 148-154.
>     - Spontaneous emotion datasets: each text corresponds to a unique emotional label where speakers naturally express emotions aligned with the semantic content.
>     - Elicited emotion datasets: speakers deliver predefined emotions with varying categories and intensities using identical texts. This approach produces multiple emotional renditions per text, where vocal expressions are intentionally elicited rather than semantically matched to the textual content.
>
> ## Re question 4:
> - Z denotes the set of integers
> - Analogous to $R^3$ representing 3D real space, $Z^1$ refers to one-dimensional integer space
> - $x_{adv}∈ Z^3$ specifies three-dimensional token values
>
> ## Re question 5:
> - The speaker embedding is extracted by a voiceprint model [[1]](https://github.com/modelscope/3D-Speaker/) specifically designed for speaker verification and diarization. This model based on ERes2Netv2 integrates both local and global acoustic features, trained on large-scale datasets containing over 10,000 speakers. It captures unique vocal characteristics that serve as distinct speaker identifiers, remaining orthogonal to emotional representations.
>
> - Such voiceprint models are widely adopted in voice cloning and TTS systems like CosyVoice and IndexTTS.
>
> - We integrate speaker embeddings in two modules:
>
>     - The LLM input, enabling it to learn emotion control orthogonal to speaker characteristics;
>
>     - The Flow model input, supplementing speech semantic tokens with timbre information during spectrogram reconstruction.
>
>     The Speaker Similarity (SS) metric in Table 1 objectively validates the model's effectiveness in cloning speaker timbre, using speaking embeddings from differently labeled synthesized speech to eliminate emotional interference.
>
> [1] https://github.com/modelscope/3D-Speaker/

---

> ### Author Response · Authors · 2025-11-14
> **Re questions 6-8**
>
> ## Re question 6:
> We have already explained this issue on Lines 201-205, 495-501, Figure 8.
> - UDDETTS's multi-task speech tokenizer utilizes Finite Scalar Quantization (FSQ) with MinMo multi-task training (inspired by CosyVoice3).
> - Unlike CosyVoice3, which is trained on multilingual inputs and acoustic events (e.g., yawns, coughs), our tokenizer focuses on precise extraction of emotional semantic information. It is jointly optimized for:
>   - speech recognition (ASR);
>   - speech emotion label recognition (SELR);
>   - speech ADV recognition (SADVR).
> - The inclusion of emotion recognition enables the intermediate codebook to capture richer paralinguistic emotional cues. By explicitly incorporating emotional modeling during quantization, our tokenizer demonstrates enhanced capacity for representing affective information in speech semantic tokens compared to CosyVoice3's approach.
> - The experimental results presented in Lines 495-501 demonstrate the emotional consistency between text semantics and synthesized speech. Figure 8 confirms that speech tokenizer effectively captures rich emotional information through the SADVR task.
> - Our multi-task speech tokenizer achieves the following performance metrics:
>
> | ASR WER(%) | SELR Accuracy(%) | SADVR RMSE (Average) | SADVR RMSE (Happy) | SADVR RMSE (Disgust)
> |   :-:      | :-:              | :-:                  | :-:                | :-:
> | 4.33       | 79.2%            | 0.68                 | 0.42               | 1.03
>
> ## Re question 7:
> - The reasons for introducing the ADV predictor and its input/output have been explained in lines 227-231.
>
>     It aims to enhance end-to-end emotional TTS by predicting ADV values solely from text input using a RoBERTa-based backbone (inspired by Park et al. [1]). It infers appropriate ADV values from textual semantics, as the end-to-end emotional TTS task requires generating emotionally appropriate speech solely from text. Thus, the ADV predictor provides textual emotional guidance, enabling UDDETTS's LLM to leverage both semantic meaning and pseudo-ADV supervision to produce speech tokens with accurate dimensional emotional representations.
> - Credibility of Pseudo-ADV Values and ADV Predictor Performance:
>
>     Figure 8 intuitively proves the accuracy and reliability of ADV mapping. Pseudo-ADV values are exclusively utilized in the end-to-end emotional TTS task. To validate the emotional consistency between text semantics (represented by pseudo-ADV values) and synthesized speech, we have supplemented the following experimental evidence in Section 4.5 (Lines 495-501) and Figure 8 (Appendix F).
>
> ## Re question 8：
> Yes, we have already open-sourced the source code, training and testing procedures, environment, and test cases on the anonymous website [[2]](https://anonymous.4open.science/w/UDDETTS/), [[3]](https://anonymous.4open.science/r/UDDETTS/), providing training support. If the paper is accepted, we will open-source more working details to facilitate user access.
>
> [1] Sungjoon Park, Jiseon Kim, et, al.. Dimensional emotion detection from categorical emotion. In Proceedings of the 2021 Conference on Empirical Methods in Natural Lang
>
> [3] https://anonymous.4open.science/w/UDDETTS/
>
> [4] https://anonymous.4open.science/r/UDDETTS/

---

> ### Author Response · Authors · 2025-11-26
> **Re weakness 5**
>
> ## Re weakness 5:
> 1. Theoretical Contributions
> - ADV Annotation Scarcity
>
>     We systematically address the challenge of scarce ADV annotations through a semi-supervised training paradigm:
>
>     - Leveraging large-scale emotional speech datasets to train the LLM;
>     - Integrating all open-source English emotional speech datasets to construct a four-category data system;
>     - Implementing a knowledge transfer mechanism from label space to ADV space, as referenced in Introduction and references (Lines 085-088, [[1]](), [[2]](), [[3]]()).
>     - In UDDETTS design, the model learns ADV-label relationships by predicting label tokens from ADV inputs. When ADV tokens are absent, the combination of text and label tokens effectively compensates for missing ADV knowledge through random masking-like techniques.
> - ADV Space Imbalance
>     Our nonlinear binning algorithm presented in Section 3.1.2 and Appendix E Table 7:
>     - Achieves optimal sample distribution across broader control units
>
>     - Precisely balances the trade-off between uniformity and discriminability
>
>     - Provides theoretical foundation for fine-grained emotion control
>
> 2. Experimental Verification
>
>     Section 4.4 validates UDDETTS' mixed emotion modeling capability from multiple perspectives:
>
> - Linear Control Verification: Table 2 and Figure 5 demonstrate linear emotion control across three dimensions
>
> - Algorithm Advantage Proof: Nonlinear binning shows higher linearity under ADV control compared to linear control algorithms
>
> - Space Coverage Improvement: Figure 4 proves UDDETTS' effectiveness in addressing ADV sparsity and imbalance while enhancing emotion space coverage
>
> - Generalization Validation: New experiments in Table 3 demonstrate UDDETTS' generalization capability for mixed emotions
>
> | Mixed emotion  | UDDETTS   | UTMOS | Similar| EmoSphere++ | UTMOS | p-value
> |    -           | :-:       | :-:   | :-:    | :-:         | :-:   | :-:
> | angry-sad      | 74.50     | 4.35  | 20.00  | 5.50        | 4.03  | 0.001
> | sleepiness-sad | 52.40     | 3.96  | 28.35  | 19.25       | 3.85  | 0.019
> | happy-surprise | 60.78     | 4.28  | 23.42  | 15.80       | 3.97  | 0.010
> | disgust-angry  | 43.33     | 4.18  | 33.34  | 23.33       | 3.90  | 0.032
>
> - Extended Verification: Appendix H further validates generalization to unseen soft/hard ADV values, with UDDETTS maintaining high ES for unseen categories (loving, anxious)
>
> These results collectively demonstrate that UDDETTS effectively solves the problems of scarcity and imbalance in ADV annotations while achieving excellent performance in fine-grained emotion control.
>
> [1] Junyu Luo, Xiao Luo, et al.. Semi-supervised fine-tuning for large language models. In Findings of the Association for Computational Linguistics: NAACL 2025, pp. 2795–2808. Association for Computational Linguistics, April 2025.
>
> [2] Juanhui Li, Sreyashi Nag, et al.. Learning with less: Knowledge distillation from large language models via unlabeled data. In Findings of the Association for Computational Linguistics: NAACL 2025, pp. 2627–2641. Association for Computational Linguistics, April 2025a.
>
> [3] Seo Yeon Park, Cornelia Caragea. VerifyMatch: A semi-supervised learning paradigm for natural language inference with confidence-aware MixUp. In Proceedings of the 2024 Conference on Empirical Methods in Natural Language Processing, pp. 19319–19335. Association for Computational Linguistics, November 2024.

---

> ### Author Response · Authors · 2025-11-28
>
> Dear Reviewer 8NAV:
>
> We have carefully revised our manuscript and incorporated suggested improvements in the updated version. We believe these revisions have significantly strengthened our work and adequately addressed your concerns. We would be grateful if you could review our responses, and would appreciate your support in increasing the final rating for our submission.
>
> We look forward to your response!

---

### Official Review · Reviewer_D694 · 2025-11-01

**Soundness:** 3
**Presentation:** 2
**Contribution:** 3
**Rating:** 4
**Confidence:** 4

**Summary:**

The paper proposes UDDETTS, a unified LLM-based framework for controllable emotional TTS that integrates both discrete emotion labels and dimensional emotions in the Arousal-Dominance-Valence (ADV) space. It addresses challenges including the sparsity and imbalance of emotional annotations, by introducing a semi-supervised training strategy and a nonlinear binning method for ADV quantization. The architecture includes a neural codec language model, an optimal-transport conditional flow matching (OT-CFM) module with an emotional mixture encoder, and a vocoder. An ADV predictor supports end-to-end synthesis from text alone. Trained on large-scale emotional and general speech datasets, UDDETTS demonstrates superior performance in label-controlled, ADV-controlled, and end-to-end TTS tasks, with linear control along ADV dimensions.

**Strengths:**

1. The paper proposes a LLM-based TTS framework to explicitly unify discrete and dimensional emotions, addressing a key limitation in prior work of emotional TTS.

2. Introducing the interpretable ADV space to LLM-based TTS is a meaningful step toward continuous, decoupled emotion control, addressing limitations of discrete-label methods. The nonlinear binning and semi-supervised fusion of annotations effectively tackle data imbalance and sparsity.

3. Evaluations across three tasks use diverse metrics (e.g., MOS, ES, SRC/KW) and show consistent improvements over baselines. The visualization in Figure 4 effectively shows that the proposed techniques (nonlinear binning, semi-supervised training) increase the coverage of the ADV space.

**Weaknesses:**

1. The novelty is limited. The work is built directly upon the architecture of models like Spark-TTS and CosyVoice. The addition of ADV control seems to be an incremental improvement rather than a novel framework.

2. The core components lack detailed explanation. For ADV quantizer, the nonlinear binning based on clustering is a potential key innovation, but its derivation and relationship to solving sparsity/imbalance are unclear in the main text.

3. The semi-supervised strategy for mixing spontaneous/elicited datasets with varying annotations is not sufficiently ablated. It's unclear if this fusion is mutually beneficial or merely a way to scale data volume.

4. The experiments are not sufficient enough. Further justifications are required.
* The baselines were not trained on the same datasets, making it difficult to determine whether the performance is influenced by model architecture or training data.
* Comparisons with other dimensional emotion models (e.g., EmoSphere++) are missing.
* The comparison against "description-based baselines" (Sec. 4.5) is potentially unfair, as these models are not designed for the specific prompt format used.
* Custom emotional texts (Table 7) lack details on design/validation for bias (e.g., inter-annotator agreement or diversity checks), risking overfitting to specific prompts.

**Questions:**

1. The nonlinear binning is central to handling sparsity. Can you provide a more intuitive explanation about how the clustering algorithm leads to a balanced and effective quantization? Why not alternatives like quantile-based or density-based methods? How sensitive is the coverage rate (89.35%) to bin count (m=14)?

2. Why is a separate RoBERTa-based regression model with MSE loss used instead of integrating ADV token prediction directly into the LLM (using CE loss like sparkTTS)? Was the above latter alternative approach explored, and if so, how did its performance compare?

3. For Sec. 4.5 comparisons, did you ensure baselines were prompted in alignment with their intended capabilities? The current setup may not fairly test them.

4. How do you demonstrate that mixing dataset types (spontaneous vs. elicited) via semi-supervised training can benefit this task effectively, rather than just adding data volume? For example, ablate training only on fully labeled data (D_{S,AL}) vs. the full setup.

---

> ### Author Response · Authors · 2025-11-17
> **Re weaknesses 1, 3, 4**
>
> ## Re weakness 1: Model Originality
> - UDDETTS employs a Transformer-based autoregressive LLM trained from scratch on ∼50k hours of data, without fine-tuning on existing models like Spark-TTS or CosyVoice.
> - We introduced many novel components absent in prior LLM-based TTS works, as shown in Lines 089-114:
>     - Integrate both ADV and label for the first time
>     - ADV space for interpretable emotion disentanglement
>     - Semi-supervised training strategy and nonlinear binning algorithm to address sparsity and imbalance issues of the ADV space
>     - An emotional mixture encoder to integrate the masked ADV tokens and label token into emotion conditions.
>     - Multi-task speech tokenizer optimized for emotional modeling
>     - ADV predictor and ADV quantizer, etc.
>
> ## Re weakness 3:
> - As referenced in Lines 515-519, we conduct an ablation study training solely on $D_{S, AL}$ without semi-supervised learning.
> - Semi-supervised learning serves two key purposes:
>   - Expands dataset coverage (↑ controllable units from 77.89% to 89.35%, Figure 4)
>   - Enables cross-dataset knowledge transfer (Lines 085–088, Table 10 in Appendix H)
>
>     UDDETTS achieves high accuracy for unseen emotion labels (e.g., loving, anxious) and generalizes well to unseen soft ADV values (Table 10), demonstrating robust interpolation in emotion space. This allows UDDETTS to recognize each label corresponds to an ADV cluster while interpreting transitional emotions (e.g., Table 6's overlapping "Unknown" regions) as natural interpolations—enabled exclusively through semi-supervised training.
>
> ## Re Weakness 4: Experiments supplement and description
> - Baseline Implementation
>
>     As referenced in Lines 907–915, all baselines are fine-tuned on the same emotional datasets.
> - Non-LLM Baseline EmoSphere++
>
>    We have supplemented the revised manuscript with a systematic comparison between UDDETTS and the traditional non-LLM-based EmoSphere++ model in Section 4.4 (Lines 465–476), Table 3, and the rightmost subfigure of Figure 4.
>    - We compare the emotional control accuracy using equivalent ADV values for four intermediate emotions: angry-sad, sleepiness-sad, happy-surprise, and disgust-angry.
>    - Reproduction & Findings:
>     EmoSphere++ is reproduced on the same emotional speech dataset.
>     Results confirm its pronounced overfitting on certain intermediate emotions. For instance, it fails to synthesize intermediate emotions along the dominance dimension between angry and sad.
>     This limitation stems from EmoSphere++'s inherent difficulty in handling limited and imbalanced ADV data [[1]]().
>
> | Mixed emotion  | UDDETTS   | UTMOS | Similar| EmoSphere++ | UTMOS | p-value
> |    -           | :-:       | :-:   | :-:    | :-:         | :-:   | :-:
> | angry-sad      | 74.50     | 4.35  | 20.00  | 5.50        | 4.03  | 0.001
> | sleepiness-sad | 52.40     | 3.96  | 28.35  | 19.25       | 3.85  | 0.019
> | happy-surprise | 60.78     | 4.28  | 23.42  | 15.80       | 3.97  | 0.010
> | disgust-angry  | 43.33     | 4.18  | 33.34  | 23.33       | 3.90  | 0.032
>
>    Justification for Initial Omission
>     - As a non-LLM-based model, EmoSphere++ exhibits noticeably inferior synthesis quality (e.g., hoarse voice), which is also observable on its official demo page [[2]](https://choddeok.github.io/EmoSphere-Demo/).
>     - EmoSphere++ adopts a spherical coordinate system (angle for emotion category, radius for intensity), which is fundamentally different from UDDETTS's Cartesian ADV control mechanism. This structural discrepancy makes direct linear ADV comparison unfeasible.
>
> - Prompt Robustness
>
> Compared SOTA models (CosyVoice2, IndexTTS2) whose original papers and demo pages [[3]](https://funaudiollm.github.io/cosyvoice2/) [[4]](https://index-tts.github.io/index-tts2.github.io/) both contain instruction examples using variable-length text descriptions (e.g., CosyVoice2's "Instructed Voice Generation" [[3]](https://funaudiollm.github.io/cosyvoice2/) and IndexTTS2's "Using Textual Description" [[4]](https://index-tts.github.io/index-tts2.github.io/)). These models demonstrate inherent robustness to diverse prompt formats, making our comparison fair.
>
> - Emotional Text Corpus
>     The revised manuscript (Lines 1096–1103; Figure 8) provides detailed descriptions and experimental results regarding the emotional text corpus:
>     - 20 neutral texts: sampled from LibriSpeech & SeedTTS, filtered via Senta [[5]](https://github.com/baidu/Senta) (≥90% neutral confidence)
>     - 10 emotional texts: GPT-5 generated, manually selected, semantically unambiguous
>     - All texts are unseen during training, eliminating overfitting concerns, with text-derived ADV predictions spanning diverse regions (see Figure 8 in Appendix F).
>
> [1] https://ieeexplore.ieee.org/stamp/stamp.jsp?tp=&arnumber=10965917
>
> [2] https://choddeok.github.io/EmoSphere-Demo/
>
> [3] https://funaudiollm.github.io/cosyvoice2/
>
> [4] https://index-tts.github.io/index-tts2.github.io/
>
> [5] https://github.com/baidu/Senta

---

> ### Author Response · Authors · 2025-11-17
> **Re weakness 2 and question 1**
>
> ## Re weakness 2 and question 1: Methodological Details
>
> - The derivation of nonlinear binning algorithm is fully detailed in Table 7 (Appendix E).
> - The nonlinear binning algorithm addresses ADV space imbalance, while semi-supervised training mitigates ADV annotation scarcity and expands the emotional datasets. Their relationship and individual roles are clarified in Lines 211–215, 085–088, 144–146.
> - Based on the detailed algorithm in Table 7 (Appendix E), the nonlinear binning is designed to group perceptually similar ADV regions into coherent emotional units, directly tackling imbalance.
> - Mechanism & Comparison: Unlike quantile-based methods (which force equal data volume per bin, risking the merge of perceptually distinct emotions in sparse regions) or pure density-based methods (which overlook perceptual structure), our approach adaptively partitions the space. It creates finer bins in dense, emotionally rich regions for precise control, while forming broader, yet acoustically consistent, bins in sparse areas to ensure robustness. This directly optimizes the trade-off between uniformity and discriminability.
>
> - Sensitivity Analysis: We added experiment in Section 4.4 (Lines 424-429) and [Figure 5](https://youke1.picui.cn/s1/2025/11/28/69287ec629642.png). The achieved coverage rate of 89.35% is robust within a range of m (12-16). The value m=14 is not arbitrarily predetermined but is algorithmically determined as the optimal cluster count K through the robust search strategy detailed in Table 6. When m<12, emotional transitions become rapid and non-smooth at extreme ADV values with reduced granularity. When m>16, insufficient samples per control unit lead to poor generalization capability and deviations in intermediate emotional transitions.
>
> - Below, we will explain the nonlinear binning algorithm in layman's terms:
>
>     - Step 1: Normalize the Data. We first scale the three emotional dimensions (arousal, dominance, valence) to a common range of [1, 7]. This ensures all dimensions contribute equally to the subsequent analysis.
>
>     - Step 2: Find the Optimal Number of Clusters (K). This step automatically determines the best number of emotional categories (K, which is the m value referenced).
>
>         (1) Initialize: Set a reasonable maximum K to explore and create a Hash table for results.
>
>         (2) Evaluate Candidates. For each candidate K (from 2 to the max, with step s), we run R times to ensure the stability of K-means. We use the silhouette score to measure the tightness of a sample with its own cluster and its separation from other clusters. A higher average silhouette score indicates a better clustering structure produced by that k-value. Store the result of the k value at each step into the hash table.
>
>         (3) Select the Best K. We first shortlist the top 25% of K values based on their average score. Then, we perform a fine-grained check on these top candidates and their immediate neighbors to avoid missing a good value. The final selection uses a formula that prioritizes both high average silhouette score and low silhouette variability across runs, ensuring a robust and stable result.
>
>     - Step 3: Cluster centers are calculated using a K-means-based clustering algorithm.
>
>     - Step 4: For bin boundaries between clusters i and i+1, we dynamically select between midpoint boundaries and weighted boundaries. The choice depends on the variance ratio rᵢ. When rᵢ > 2 (indicating significantly different distributions), we use weighted boundaries to account for cluster density differences.
>
>     - Step 5: Divide the ADV space into bins.

---

> ### Author Response · Authors · 2025-11-20
> **Re question 2, 3, 4**
>
> ## Re question 2: ADV Predictor Design
>
> - During initial development, we explored integrating ADV prediction directly into the LLM (as in our open-sourced codes) but observed higher prediction errors compared to the specialized RoBERTa-based ADV predictor.
>
> - The RoBERTa-based model provides superior accuracy in capturing nuanced semantic-emotion relationships from text, aligning with prior verified work [[1]]().
>
> | Method  | RMSE
> |    -           | :-:
> | RoBERTa-based ADV predictor   | 1.25
> | LLM integrated ADV prediction | 2.87
>
> - This modular design ensures more precise emotional control while simplifying the LLM's learning objective.
>
>     - Robust ADV annotation via a dedicated "ADV annotator"
>
>     - LLM focus on its core task: generating speech that accurately matches given ADV values
>
> ## Re question 3:
>
> - We have addressed this question in our response to the weakness 4 (3).
> - Furthermore, the core objective of this section is to demonstrate UDDETTS's unique capability to synthesize nuanced emotional speech directly from text by replacing lengthy textual prompts with dimensional emotion vectors (ADV). This represents a paradigm shift—enabling emotion synthesis through structured space navigation rather than linguistic description interpretation. This ability to capture the inner emotions of a text can be found in [Figure 8](https://youke1.picui.cn/s1/2025/11/28/6928828099ff2.png) in Appendix F.
>
>
> Represents a paradigm shift: enabling emotion synthesis through structured space navigation rather than linguistic description interpretation.
>
> ## Re question 4:
>
> - We have addressed this question in our response to the weakness 3.
>
>   - As referenced in Lines 515-519, we conduct an ablation study training solely on $D_{S, AL}$ without semi-supervised learning.
>   - Refer to Figure 4, Lines 085–088, 471-476, and Table 9, 10 in Appendix H.
>
>     Training only on $D_{S,AL}$ leaves large ADV regions uncontrollable, with poor robustness and unstable vocal expressions in under-generalized areas due to insufficient annotation breadth.
>
> [1] SungjoonPark, JiseonKim, et.al.. Dimensional emotion detection from categorical emotion. In Proceedings of the 2021 Conference on Empirical Methods in Natural Language Processing, pp.4367–4380. Association for Computational Linguistics, November 2021.

---

> ### Author Response · Authors · 2025-11-28
>
> Dear Reviewer D694:
>
> We have carefully revised our manuscript and incorporated suggested improvements in the updated version. We believe these revisions have significantly strengthened our work and adequately addressed your concerns. We would be grateful if you could review our responses, and would appreciate your support in increasing the final rating for our submission.
>
> We look forward to your response!

---

### Official Review · Reviewer_bqm5 · 2025-11-01

**Soundness:** 3
**Presentation:** 3
**Contribution:** 3
**Rating:** 6
**Confidence:** 4

**Summary:**

This paper proposes UDDETTS, a LLM framework that unifies discrete and dimensional emotions for controllable emotional text-to-speech (TTS). The framework introduces an interpretable ADV space to describe dimensional emotions, supporting emotion control driven by discrete emotion labels or non-linearly quantized ADV values. Moreover, this paper designs asemi-supervised training strategy to fully utilize speech datasets with different emotion annotation types, experimental results show promising emotion controlablety for speech synthesis.

**Strengths:**

1. The paper tackles a critical and timely problem in expressive speech synthesis, contious and dimensional control is a clear and important direction for the field.
2. The semi-supervised learning strategy is an effective solution to extend the training to larger-scale dataset, while only part of the data is well labeled.

**Weaknesses:**

1. Although this article compares many different baselines, the reasonableness of the comparison is still not clear to me. A more reasonable comparison would be to add the adv prediction and control modules to the corresponding frameworks, which would better illustrate the universality of the article's contribution.
2. Some details are not very clear. For example, in Table 3, preference scores are given for two systems, but it is uncertain whether the same backbone is used for the corresponding systems, and it is also uncertain whether the baseline systems have been optimized with similar training methods using the same emotional data.
3. The article mentions some other control schemes, such as EmoSphere-TTS, but they are not shown in the experimental results.

**Questions:**

Besides the issues mentioned in the weakness,

1. How sensitive is the model's performance to the ratio of ADV-annotated data versus label-only data?
2. Has the accuracy of the ADV predictor been tested standalone?
3. Was any experiment conducted where the ADV predictor and the emotional mixture encoder were integrated into a different LLM-based framework, such as IndexTTS2 or Spark-TTS, to measure the performance lift?

**Details Of Ethics Concerns:**

The Ethics issue on generatative models should be discussed in more detail.

---

> ### Author Response · Authors · 2025-11-20
> **Re weaknesses and questions 3**
>
> ## Re weakness 1 and question 3:
>
> 1. Fair Comparison Assurance
>
>     Architectural Consistency: Both UDDETTS and all baseline models adopt LLM-based architectures with:
>     - Comparable parameter counts
>     - Similar training dataset scale
>
> 2. Experimental Validation
>
>     - Our initial UDDETTS prototype was developed by restructuring and retraining the CosyVoice model from the CosyVoice-300M checkpoint.
>     - The current version of UDDETTS has been further refined and retrained from scratch without relying on any pre-existing LLM checkpoints. As specified in Appendix C (Lines 907–915), all baseline models were retrained on the same emotional speech datasets to ensure an equitable comparison. It is widely recognized in methodological practice that novel architectures are evaluated against retrained baselines, rather than re-implementing core innovations across all comparator models.
>     - In the revised manuscript, we have added a comparison between CosyVoice and CosyVoice + ADV in Section 4.3 (Lines 373–402) and Table 1.
>
> | Models        | MOS       | $P_m$ | $R_m$ | UTMOS | WER(%)| SS    | ES    | STOI  | PESQ-WB
> |    -          | :-:       | :-:   | :-:   | :-:   | :-:   | :-:   | :-:   | :-:   | :-:
> | CosyVoice     | 4.02±0.08 | 0.83  | 0.73  | 3.87  | 4.35  | 0.679 | 0.635 | 0.83  | 2.16
> | CosyVoice+ADV | 4.15±0.05 | 0.90  | 0.81  | 4.10  | 4.08  | 0.680 | 0.815 | 0.86  | 2.66
>
> 3. Experimental Findings
>
>    Comparative results confirm that the ADV mechanism substantially enhances both synthesis quality and emotional control precision, validating our architectural innovations while maintaining fair comparison standards.
>
> ## Re weakness 2:
>
> 1. Implementation Transparency
>
>     All baseline implementation details are fully documented in Appendix C (Lines 907–915).
>     We strictly follow official implementations from respective papers to ensure reproducibility
>
> 2. Ensure Fairness
>
>    To ensure fairness, all LLM-based baselines with publicly available pretrained checkpoints and codes are fine-tuned for 10 epochs until convergence solely on our emotional speech datasets, using label prompts as training inputs (e.g., "Angry[object Object]ContentText").
>
> 3. Architectural Consistency
>    CosyVoice2 and IndexTTS2, which are compared in Table 3, are also based on Transformer LLM as the backbone for synthesizing speech tokens. Their descriptions are listed in Appendix C.
>
> ## Re weakness 3:
>
> 1. Additional Comparative Experiments
>
>    We have supplemented the revised manuscript with a systematic comparison between UDDETTS and the traditional non-LLM-based EmoSphere++ model in Section 4.4 (Lines 465–476), Table 3, and the rightmost subfigure of Figure 4.
>    - We compare the emotional control accuracy using equivalent ADV values for four intermediate emotions: angry-sad, sleepiness-sad, happy-surprise, and disgust-angry.
>    - Reproduction & Findings:
>     EmoSphere++ is reproduced on the same emotional speech dataset.
>     Results confirm its pronounced overfitting on certain intermediate emotions. For instance, it fails to synthesize intermediate emotions along the dominance dimension between angry and sad.
>     This limitation stems from EmoSphere++'s inherent difficulty in handling limited and imbalanced ADV data (Lines 138–141, Figure 4, [[1]](https://ieeexplore.ieee.org/stamp/stamp.jsp?tp=&arnumber=10965917)).
>
> | Mixed emotion  | UDDETTS   | UTMOS | Similar| EmoSphere++ | UTMOS | p-value
> |    -           | :-:       | :-:   | :-:    | :-:         | :-:   | :-:
> | angry-sad      | 74.50     | 4.35  | 20.00  | 5.50        | 4.03  | 0.001
> | sleepiness-sad | 52.40     | 3.96  | 28.35  | 19.25       | 3.85  | 0.019
> | happy-surprise | 60.78     | 4.28  | 23.42  | 15.80       | 3.97  | 0.010
> | disgust-angry  | 43.33     | 4.18  | 33.34  | 23.33       | 3.90  | 0.032
>
> 2. Justification for Initial Omission
>     - As a non-LLM-based model, EmoSphere++ exhibits noticeably inferior synthesis quality (e.g., hoarse voice), which is also observable on its official demo page [[2]](https://choddeok.github.io/EmoSphere-Demo/).
>     - EmoSphere++ adopts a spherical coordinate system (angle for emotion category, radius for intensity), which is fundamentally different from UDDETTS's Cartesian ADV control mechanism. This structural discrepancy makes direct linear ADV comparison unfeasible.
>
> [1] https://ieeexplore.ieee.org/stamp/stamp.jsp?tp=&arnumber=10965917
>
> [2] https://choddeok.github.io/EmoSphere-Demo/

---

> ### Author Response · Authors · 2025-11-25
> **Re questions 1-2**
>
> ## Re question 1:
>
> - As referenced in Lines 515-519, we conduct an ablation study training solely on $D_{S, AL}$ without semi-supervised learning.
>
> - We further investigate the impact of ADV-annotated data quantity by halving the proportion of ADV-annotated data relative to label-only data during second-stage training.
>
>   Performance impact:
>   - On the label-controlled emotional TTS task, the objective Emotion Similarity (ES) metric decreases from 0.833 to 0.805.
>   - On the ADV-controlled emotional TTS task, no significant difference is observed in the subjective evaluations of linear emotion control along the three ADV dimensions, demonstrating that fine-grained linear control remains feasible across these decoupled dimensions.
>   - The controllable coverage rate declines to 78.95% with a corresponding reduction in the number of samples per controllable unit.
>   - The MOS score for soft ADV values, e.g., [3,4,10] in Table 10, from 4.08 drops to 3.60.
>   - The normality of the ADV probability distribution in Figure 7 is compromised, manifesting as a bimodal distribution emerging along the Arousal dimension.
>
> ## Re question 2:
>
> - ADV Predictor Performance: Our results in Lines 496-497 show the ADV predictor achieves an RMSE of 1.25, slightly higher than the multi-task speech tokenizer's RMSE of 0.68 on the SADVR task.
>
> - Cross-modal Consistency: While textual ambiguity causes minor misalignment between textual and vocal emotion representations, the deviation remains perceptually subtle. This is supported by the close alignment between text-derived and speech-derived ADV values shown in Lines 495-501 and Figure 8.

---

### Author Response · Authors · 2025-11-26
**To All Reviewers**

Thank you very much for your valuable comments and suggestions. We have made our best efforts to address all questions raised in the reviews and have supplemented the manuscript with relevant experiments.

We sincerely hope you will actively engage with our rebuttal responses and consider our other comments and replies to better understand our work. We would greatly appreciate your recognition of our research contributions.

All experimental results have been incorporated into the revised manuscript, which has been updated to meet the 10-page limit in accordance with ICLR requirements. A summary of all revisions has been prepared and included in the supplementary materials for your reference.

We look forward to your continued feedback and hope our revisions adequately address your concerns.

---

### Meta-Review · Area_Chair_m9qY · 2026-01-07

**Summary:**

Reviewers agreed that the paper targets a relevant and timely problem—controllable emotional TTS that unifies discrete labels and continuous ADV representations—and recognized the scope of the proposed framework and experiments. However, several concerns informed the decision.

First, reviewers noted insufficient clarity and justification of core design choices, including the nonlinear ADV binning strategy, the semi-supervised training scheme, and the interaction between ADV-based and label-based emotion control. Questions were raised regarding robustness, sensitivity to hyperparameters, and behavior under conflicting emotion signals.

Second, there were concerns about data reliability, as the amount of ground-truth ADV-labeled data is limited and the system relies heavily on pseudo-ADV values inferred through semi-supervised learning. Reviewers questioned whether this reliance may affect the accuracy and interpretability of fine-grained emotional control, particularly for subtle or mixed emotions.

Third, while the experimental evaluation is extensive, some reviewers expressed reservations about the strength and fairness of the empirical evidence, including baseline comparisons and perceptual evaluations, and felt that additional analyses would be necessary to fully substantiate the claimed advantages.

Overall, reviewer opinions varied, reflecting both the paper’s promise and remaining uncertainty about the robustness and generality of its contributions, which ultimately influenced the suggested decision.

**Reviewer Concerns:**

Concerns addressed by the rebuttal:
The rebuttal effectively clarified several methodological issues raised by reviewers. In particular, it provided detailed explanations and additional experiments addressing (1) the nonlinear ADV binning strategy, including sensitivity analysis on bin count and robustness across splits; (2) the role and necessity of semi-supervised learning in mitigating ADV annotation sparsity; and (3) the interaction between ADV-based and label-based emotion control, supported by additional analyses on ADV–label alignment and conflict robustness. The authors also strengthened the experimental section by adding fairer baseline comparisons, supplementary perceptual evaluations, and clearer documentation of datasets, evaluation protocols, and implementation details.

Concerns still outstanding:
Despite these improvements, some concerns remain only partially resolved. Notably, the reliance on pseudo-ADV values due to limited ground-truth ADV annotations continues to raise questions about the ultimate precision and reliability of fine-grained continuous emotion control, especially for subtle or compound emotions. Additionally, while robustness analyses were added, long-term generalization and interpretability of the learned ADV representations across broader datasets and settings remain somewhat uncertain. Finally, although experimental coverage was expanded, some reviewers may still view the empirical evidence as insufficient to fully eliminate doubts about the consistency and dominance of emotion signals under all conditions.

Overall, the rebuttal substantially improved clarity and addressed many technical questions, but a degree of uncertainty remains regarding data limitations and the generality of the proposed approach.

**Reviewer Scores:**

Reviewer bqm5 (6): Most concerns were addressed in the rebuttal, so the score would likely stay the same or go up slightly.

Reviewer D694 (4): With the added explanations and experiments, the score would likely increase to around the borderline accept range.

Reviewer 8NAV (2): Even though many points were addressed, the score would probably increase only a little.

Reviewer 2DR5 (6): The main concerns were clearly answered, so the score would likely stay the same or increase slightly.

---

### Decision · Program_Chairs · 2026-01-26

Reject